# Detecting and tracking eddies in oceanic flow fields: A Lagrangian descriptor based on the modulus of vorticity

Rahel Vortmeyer-Kley[1], Ulf Gräwe[2, 3], and Ulrike Feudel[1]

[1]Institute for Chemistry and Biology of the Marine Environment, Theoretical Physics/Complex Systems, Carl von Ossietzky University Oldenburg, Oldenburg, Germany
[2]Leibniz Institute for Baltic Sea Research, Rostock-Warnemünde, Germany
[3]Institute of Meteorology and Climatology, Leibniz University Hannover, Hannover, Germany

*Correspondence to:* Rahel Vortmeyer-Kley (rahel.vortmeyer-kley@uni-oldenburg.de)

**Abstract.** Since eddies play a major role in the dynamics of oceanic flows, it is of great interest to detect them and gain information about their tracks, their lifetimes and their shapes. We present a Lagrangian descriptor based on the modulus of vorticity to construct an eddy tracking tool. In our approach we denote an eddy as a rotating region in the flow possessing an eddy core corresponding to a local maximum of the Lagrangian descriptor and enclosed by pieces of manifolds of distinguished hyperbolic trajectories (eddy boundary). We test the performance of the eddy tracking tool based on this Lagrangian descriptor using an convection flow of four eddies, a synthetic vortex street and a velocity field of the western Baltic Sea. The results for eddy lifetime and eddy shape are compared to the results obtained with the Okubo-Weiss parameter, the modulus of vorticity and an eddy tracking tool used in oceanography. We show that the vorticity based Lagrangian descriptor estimates lifetimes closer to the analytical results than any other method. Furthermore we demonstrate that eddy tracking based on this descriptor is robust with respect to certain types of noise which makes it a suitable method for eddy detection in velocity fields obtained from observation.

## 1   Introduction

Transport of particles and chemical substances mediated by hydrodynamic flows are important components in the dynamics of ocean and atmosphere. For this reason, there is an increasing interest in identifying particular structures in the flows such as eddies or transport barriers to understand their role in transport and mixing of the fluid as well as their impact on marine biology for instance. Of particular interest in oceanography are eddies, which can be responsible for the confinement of plankton within them and hence, important for the development of plankton blooms (Abraham (1998); Martin et al. (2002); Sandulescu et al. (2007)). Such eddies possess a large variety of sizes and lifetimes. To tackle the problem of recognizing such eddies in aperiodic flows, different approaches have been developed: on the one hand, there are several methods available which are inspired by dynamical systems theory (Haller (2015); Mancho et al. (2013) and references therein), on the other hand, numerical software for automated eddy detection has been developed in oceanography based on either physical quantities of the flow (Okubo (1970); Weiss (1991); Nencioli et al. (2010)) or geometric measures (Sadarjoen and Post (2000)).

Algorithms for finding eddies in fluid flows are applied in very different fields of science such as in atmospheric science (Koh

and Legras (2002)), celestial mechanics (Gawlik et al. (2009)), biological oceanography (Bastine and Feudel (2010); Huhn et al. (2012)) and the dynamics of swimmers (Wilson et al. (2009)). The largest field of application is oceanography, since oceanic flows contain a large number of mesoscale eddies of size 100-200 km, which are important components of advective transport. Their emergence and lifetime influences the transport of pollutants (Mezić et al. (2010); Olascoaga and Haller (2012); Tang and Luna (2013)) or plankton blooms (Bracco et al. (2000); Sandulescu et al. (2007); Rossi et al. (2008); Hernández-Carrasco et al. (2014)). There is an increasing number of eddy resolving data sets available provided either by observations (Donlon et al. (2012)) or by numerical simulations (Thacker et al. (2004); Dong et al. (2009)). Subsequently there is a growing interest in the census of eddies, their size and lifetimes depending on the season. This task requires robust algorithms for the computation of eddy boundaries as well as the precise detection of their appearance and disappearance in time based on numerical velocity fields (Petersen et al. (2013); Wischgoll and Scheuermann (2001); Dong et al. (2014)) as well as altimetry data (Chaigneau et al. (2008); Chelton et al. (2011)). However, the huge amount of available data poses a challenge to data analysis. As pointed out in Chaigneau et al. (2008) mesoscale and submesoscale eddies cannot be extracted from a turbulent flow without a suitable definition and a competitive automatic identification algorithm. Several such algorithms have been developed based on the various concepts mentioned above. In the following we will briefly discuss several of those algorithms.

Based on dynamical systems theory, one can search for Lagrangian coherent structures (LCS) which describe the most repelling or attracting manifolds in a flow (Haller and Yuan (2000)). The time evolution of these invariant manifolds make up the Lagrangian skeleton for the transport of particles in fluid flows. LCS can be considered as the organizing centres of hydrodynamic flows. Their computation is based on the search for stationary curves of shear in case of hyperbolic or parabolic LCS. Elliptic LCS like eddies are computed as stationary curves of averaged strain (Haller and Beron-Vera (2013); Karrasch et al. (2015); Onu et al. (2015)) or Lagrangian-averaged vorticity deviation (Haller et al. (2016)). Other methods to determine whether an eddy can be identified in the flow employ average Lagrangian velocities (Mezić et al. (2010)) or burning invariant manifolds (Mitchell and Mahoney (2012)). The latter have been introduced originally to track fronts in reaction diffusion systems (Mahoney et al. (2012)) but have been recently extended to the detection of eddies (Mahoney and Mitchell (2015)). A completely different approach which connects geometric properties of a flow with probabilistic measures utilizes transfer operators to identify LCS (Froyland and Padberg (2009)). Another approach is the computation of distinguished hyperbolic trajectories (DHT) and their stable and unstable manifolds to identify Lagrangian coherent structures in a flow. DHTs can be considered as a generalization of stagnation points of saddle type and their separatrices to general time-dependent flows (Ide et al. (2002); Wiggins (2005); Mancho et al. (2006)). DHTs and their manifolds can be computed using Lagrangian descriptors, which integrate intrinsic physical properties for a finite time and thereby reveal the geometric structures in phase space (Mancho et al. (2013)). Stable and unstable manifolds can also be calculated using the ridges of finite time or finite size Lyapunov exponents (FTLE or FSLE) (Artale et al. (1997); Boffetta et al. (2001); d'Ovidio et al. (2004); Branicki and Wiggins (2010)) using the idea that initially nearby particles in a flow will move apart in stretching regions while they will move closer to each other in contracting regions.

Despite the discussion about objectivity (cf. Haller's short comment SC2 in the discussion of this paper, Mancho's editor comment EC1 and Mendoza and Mancho (2012)) the method of Lagrangian descriptors is very appealing and is appropriate to

gain insight into oceanographic flows. It has already been successfully applied to compute Lagrangian coherent structures in the Kuroshio current (Mendoza et al. (2010); Mendoza and Mancho (2010, 2012)), in the Polar Vortex (de la Cámara et al. (2012)), in the North-Western Mediterranean Sea (Branicki et al. (2011)) as well as analysing the possible dispersion of debris from the Malaysian Airlines flight MH370 airplane in the Indian Ocean (García-Garrido et al. (2015)).

In the recent years there has been some effort to derive Eulerian quantities which can be used to draw conclusions about Lagrangian transport phenomena (Sturman and Wiggins (2009); McIlhany et al. (2011); McIlhany and Wiggins (2012); McIlhany et al. (2015)).

    In oceanography, one of the most popular methods to identify eddies is based on the Okubo-Weiss parameter (Okubo (1970); Weiss (1991)). This method relies on the strain and vorticity of the velocity field, and has been applied to both, numerical

ocean model output and satellite data (Isern-Fontanet et al. (2006); Chelton et al. (2011)). Often, the underlying velocity field is derived from altimetric data under the assumption of geostrophic theory. In this approach two limitations can appear. First, the derivation of the velocity field can induce noise in the strain and vorticity field. This is usually reduced by applying a smoothing algorithm, which might, in turn, remove physical information. Secondly, Douglass and Richman (2015) show that eddies can have a significant ageostrophic contribution. Thus, the detection might fail when relying on geostrophic theory. A

slightly different approach was developed by Yang et al. (2001) and Fernandes et al. (2011), who used the signature of eddies in the sea surface temperature (SST) to detect them. The partially sparse coverage of satellite SST data limits the application of this method.

    Sadarjoen and Post (2000) developed a tracking algorithm that is based on the flow geometry. The assumption is that eddies can be defined as features characterized by circular or spiral streamlines around the core of an eddy. The streamlines are derived

from the velocity field. Additionally, the change of direction of the segments that compose the streamline (winding angle) is computed for each streamline. Chaigneau et al. (2008) applied this winding angle approach to a data set of the South Pacific. Moreover, they compared the winding angel method to the Okubo-Weiss approach and concluded that the former is more successful in detecting eddies and more important with a much smaller excess of detection errors. A further method based on geometric properties is proposed by Nencioli et al. (2010). The underlying idea is that within an eddy, the velocity field

changes its direction in a unique way. Moreover, the relative velocity in the eddy core should vanish and should be enclosed by closed stream lines. This detection and tracking algorithm was successfully applied by Dong et al. (2012) in the Southern California Bight. In addition, the detection algorithm of Nencioli et al. (2010) has the advantage that its application is not limited to surface fields (Isern-Fontanet et al. (2006); Chelton et al. (2011); Fernandes et al. (2011)). Thus, it is possible to track eddies in the interior of the ocean, without any surface signature.

In this paper we develop an eddy detection and tracking tool based on the method of the Lagrangian descriptor introduced by Mancho and co-workers (Madrid and Mancho (2009); Mancho et al. (2013)). For the purpose of automated eddy detection we propose to use the modulus of the vorticity as the scalar quantity to be computed along a trajectory instead of using the arc length of trajectories. We compare our method to four others, namely the original Lagrangian descriptor using the arc length (Madrid and Mancho (2009); Mendoza et al. (2010)), an oceanographic method based on geometric properties of the flow field

(Nencioli et al. (2010)) and detection tools which employ the Okubo-Weiss parameter (Okubo (1970); Weiss (1991)) and the

vorticity itself.

The paper is organized as follows: Sect. 2 briefly reviews the Eulerian concepts vorticity and Okubo-Weiss parameter, the Lagrangian descriptors $M$ based on the arc length and $M_V$ based on the modulus of vorticity. To compare the performance of the two Lagrangian descriptors and the Eulerian concepts we use two simple velocity fields: the model of four counter rotating
eddies and a modified van Kármán vortex street in Sect. 3. In Sect. 4 we describe the implementation of the Lagrangian descriptor based on the modulus of vorticity as a tracking tool identifying eddy lifetimes (Sect. 4.1) and compare the results again with the aforementioned other methods. In Sect. 4.2 we study the performance of the method in cases where we corroborate the velocity fields with noise to test the robustness of the method if applied to velocity fields obtained from observational data. Finally in Sect. 4.3 we compare the Eulerian and the Lagrangian view of the eddy shape with application to the modified van
Kármán vortex street and to a velocity field from oceanography describing the dynamics of the Western Baltic Sea (Gräwe et al. (2015a)). We conclude the paper with a discussion in Sect.5.

## 2  From Eulerian to Lagrangian methods

The dynamics of a fluid can be characterized employing two different concepts: the Eulerian and the Lagrangian view. While the Eulerian view uses quantities describing different properties of the velocity field, the Lagrangian view provides quantities
from the perspective of a moving fluid particle. Out of the variety of different Eulerian and Lagrangian methods mentioned in the introduction, we recall here briefly only those concepts, which will be important for our development of a measure to identify eddies in a flow.

An Eulerian method to describe the circulation density of a velocity field in hydrodynamics is vorticity $\boldsymbol{W}(\boldsymbol{x}, t)$ defined as the curl of the velocity field $\boldsymbol{v}(\boldsymbol{x}, t)$. The vorticity associates a vector to each point in the fluid representing the local axis of rotation
of a fluid particle. It displays areas with a large circulation density like eddies as regions of large vorticity and eddy cores as local maxima.

Another Eulerian quantity is the Okubo-Weiss parameter $OW$. It weights the strain properties of the flow against the vorticity properties and distinguishes so strain dominated areas from vorticity dominated one. The Okubo-Weiss parameter is defined as

$$OW = s_n^2 + s_s^2 - \omega^2, \tag{1}$$

where the normal strain component $s_n$, the shear strain component $s_s$ and the relative vorticity $\omega$ of a two dimensional velocity field $\boldsymbol{v} = (u, v)$ are defined as

$$s_n = \frac{\partial u}{\partial x} - \frac{\partial v}{\partial y}, \;\; s_s = \frac{\partial v}{\partial x} + \frac{\partial u}{\partial y} \;\text{ and }\; \omega = \frac{\partial v}{\partial x} - \frac{\partial u}{\partial y}. \tag{2}$$

Eddies are areas having a negative Okubo-Weiss parameter with a local minimum at the eddy core because here the vorticity
component outweighs the strain component, while strain dominated areas are characterized by a positive Okubo-Weiss parameter.

A Lagrangian view on the dynamics of the velocity field is given by the Lagrangian descriptor developed by Mancho and coworkers (Madrid and Mancho (2009)). A more general definition of the Lagrangian descriptor is outlined in Mancho et al. (2013). Here we focus on the Lagrangian descriptor based on the arc length of a trajectory, defined as

$$
M(\boldsymbol{x}^*, t^*)_{\boldsymbol{v}, \tau} = \int_{t^*-\tau}^{t^*+\tau} \left( \sum_{i=1}^{n} \left( \frac{dx_i(t)}{dt} \right)^2 \right)^{1/2} dt \tag{3}
$$

with $\boldsymbol{x}(t) = (x_1(t), x_2(t)...x_n(t))$ being the trajectory of a fluid particle in the velocity field $\boldsymbol{v}$ that is defined in the time interval $[t^* - \tau, \ t^* + \tau]$ and going through the point $\boldsymbol{x}^*$ at time $t^*$.

The Lagrangian descriptor $M$ yields singular features that can be interpreted as time-dependent "phase space structures" like (time-dependent or moving) elliptic or hyperbolic "fixed" points (denoted as distinguished elliptic or hyperbolic trajectories DET and DHT respectively in Madrid and Mancho (2009)) and their time dependent stable and unstable manifolds (Mancho

et al. (2013); Wiggins and Mancho (2014)). The reason for the singular features of $M$ is, that $M$ accumulates different values of the arc length depending on the dynamics in the region. Trajectories that have a similar dynamical evolution yield similar values of $M$. When the dynamics changes abruptly, $M$ will change too. This is the case at distinguished hyperbolic trajectories (DHTs) and their stable and unstable manifolds. Trajectories on both sides of the manifold have a different behaviour compared to the behaviour of the trajectories on the manifold. Either they approach the manifold very fast or they move away from the

manifold very fast. In both cases they accumulate larger values of $M$ in a given time interval than trajectories on the manifold. Therefore, the singular line of $M$ in a color-coded plot of $M$ can be interpreted as corresponding to a manifold. If a trajectory stays in a region or at one point it accumulates a small or zero value of $M$ and $M$ becomes a local minimum. While DHTs have been extensively studied, distinguished trajectories possessing an elliptic type are less understood. However, such trajectories can also be identified as singular features of $M$ being surrounded by an elliptic region in the sense of Mancho et al. (2013). For

an extensive discussion about the notion of hyperbolic and elliptic regions in flows we refer to Mancho et al. (2013).

For each instant of time $t^*$ the color-coded plots of $M$ can be interpreted as a "snapshot" of the phase space, where the minima correspond to one point of a DHT or a distinguished trajectory surrounded by an elliptic region. When $t^*$ is changing $M$ reveals the time evolution of the phase space and loosely speaking distinguished hyperbolic trajectories can be considered as "moving saddle points", distinguished trajectories surrounded by an elliptic region in the sense of Mancho et al. (2013) as "moving

elliptic points". Due to the arbitrary time-dependence of the flow, both, the DHTs and the distinguished trajectories surrounded by an elliptic region are time-dependent and exist in general only for a finite time in a time-dependent flow. Hyperbolicity in case of DHT refers to the fact that along those trajectories Lyapunov exponents are positive or negative, but not zero except for the direction along the trajectory (Mancho et al. (2013)).

Because the Lagrangian descriptor $M$ would display minima in both cases, i.e. DHT and distinguished trajectories surrounded

by an elliptic region, a second criterion is needed to distinguish them properly. To avoid such additional distinction criterion, we suggest a Lagrangian descriptor based on the modulus of vorticity to simplify the automated eddy detection. We emphasize, that it has already been pointed out by Mancho et al. (2013) that any intrinsic physical or geometrical property of trajectories can be used to construct a Lagrangian descriptor by integrating this property along trajectories over a certain time interval.

Therefore, we introduce a vorticity based Lagrangian descriptor $M_V$ in which the physical quantity is the modulus of the vorticity $W$ of a velocity field $\boldsymbol{v}(\boldsymbol{x}, t)$

$$W(\boldsymbol{x}, t) = |\nabla \times \boldsymbol{v}(\boldsymbol{x}, t)|. \tag{4}$$

We define the Lagrangian descriptor $M_V$ based on the modulus of vorticity as

$$M_V(\boldsymbol{x}^*, t^*)_\tau = \int\limits_{t^*-\tau}^{t^*+\tau} (W(\boldsymbol{x}, t))^{1/2} dt. \tag{5}$$

The Lagrangian descriptor $M_V$ based on the modulus of vorticity measures the Eulerian quantity modulus of vorticity along a trajectory (Lagrangian view) passing through a position $\boldsymbol{x}^*$ at time $t^*$ in a time interval $[t^* - \tau, \ t^* + \tau]$. Within this time interval trajectories accumulate different values of $M_V$. As the arc length based Lagrangian descriptor $M$, the Lagrangian descriptor $M_V$ based on the modulus of vorticity displays singular features as lines or local minima or maxima. In case of

local maxima, a trajectory does not leave the region of large values of modulus of vorticity. Such regions are typical for the inner part of an eddy. Therefore, a local maximum corresponds to the eddy core and can be interpreted as a "snapshot" of the distinguished trajectory at time $t^*$ surrounded by an elliptic region in the sense of Mancho et al. (2013). By contrast, local minima of $M_V$ arise if a trajectory stays in a region of small values of modulus of vorticity. In analogy with the singular lines in case of $M$, singular lines of $M_V$ can be interpreted as the boundaries of regions of different dynamical behaviour. In this

sense they can be understood as manifolds of the DHTs.

The local maxima and the singular lines of $M_V$ will be used to construct an eddy tracking tool based on the following concept of an eddy: We denote an eddy as being bounded by pieces of stable and unstable manifolds of DHTs (according to Branicki et al. (2011) and Mendoza and Mancho (2012)) surrounding an area in which the flow is rotating. The manifolds correspond to singular lines in $M_V$ which are used to describe the eddy boundaries. The eddy core is considered to be a local maximum

of $M_V$ within this bounded region and can be interpreted as one point of a distinguished trajectory surrounded by an elliptic region.

In case of $M_V$ as well as in case of $M$ the resolution of these structures depends on the choice of the parameter $\tau$ that gives the length of the time interval. Structures that live shorter than $2\tau$ cannot be resolved. Even structures that live longer than $2\tau$ can only be resolved if $\tau$ is chosen large enough. The choice of $\tau$ depends on the structure and the time scale of the flow field

considered. Within the range of the time scale of the problem that should be resolved some variation of $\tau$ is needed until the optimal $\tau$ for a given problem is found.

## 3 Eddies in a flow: Comparing Eulerian and Lagrangian methods

To compare the performance of the proposed Lagrangian descriptor based on the modulus of vorticity to the others, two test cases - a convection flow consisting of four counter rotating eddies and a model of a vortex street - are used. The four counter

rotating eddies are employed to show that different methods detect different aspects of the eddies. Additionally, we discuss

how the displayed structure depends on the chosen $\tau$. The vortex street is particularly used to test how suitable our method is to detect and track eddies in comparison to other methods and how well they all estimate eddy lifetimes and shapes. This way we gain insight into performance, advantages and disadvantages of the proposed method compared to the others.

To give a complete view of the advantages and disadvantages the results of the different test cases are interpreted in a coherent
discussion after presenting all results.

The equations of motion of fluid particles in a convection flow of four counter rotating eddies are given by

$$u = \dot{x} = \sin(2\pi \cdot x) \cdot \cos(2\pi \cdot y) \ \text{and} \ v = \dot{y} = -\cos(2\pi \cdot x) \cdot \sin(2\pi \cdot y). \tag{6}$$

We compute the four different quantities, modulus of vorticity, Okubo-Weiss parameter, the two Lagrangian descriptors $M$ and $M_V$ on a spatial domain $[0, 1] \times [0, 1]$. To this end, the spatial domain is decomposed into a discrete grid $(201 \times 201)$ and the
different methods are calculated for each grid point. The results are presented in Figs. 1 and 2.

The model of the vortex street consists of two eddies that emerge at two given positions in space, travel a distance $L$ in positive $x$-direction and fade out. The two eddies are counter rotating. They emerge and die out periodically with a time shift of half a period. The model is adapted from Jung et al. (1993) and Sandulescu et al. (2006) with the difference that the cylinder as the cause of eddy formation and its impact on the flow field due to its shade is neglected. In this sense, the eddies emerge
non-physically out of nowhere, but all quantities like lifetime and radius to be estimated by means of eddy tracking are then given analytically and make up a perfect test scenario. A detailed description of the model can be found in the supplemental material to this article. Again all methods are applied to this velocity field using a $(302 \times 122)$ grid. Unless otherwise stated, the time interval $\tau$ for the Lagrangian methods is set to $0.15$ times the lifetime of an eddy. The results are presented in Fig. 3.

These two test cases reveal the following characteristics of the properties of coherent structures in a flow: Eulerian as well
as Lagrangian methods display eddy cores as local maxima (modulus of vorticity, $M_V$) or local minima (Okubo-Weiss, $M$) of the respective quantity (Fig.1,2,3). Local minima of the Lagrangian methods correspond to DHTs (Fig. 2 (e), (f)). For the Lagrangian descriptor $M$ the core of the eddy as well as the DHT are indistinguishable since they are both displayed as local minima of $M$. The Lagrangian descriptor $M_V$ based on modulus of vorticity can clearly distinguish between the core of an eddy and a DHT (Fig. 2(a)-(c)). For this reason, Eulerian methods and the Lagrangian descriptor $M_V$ are more appropriate
than the Lagrangian descriptor $M$ for an automated identification of eddies, since no further criteria are needed.

To characterize Lagrangian coherent structures in a flow not only distinguished trajectories surrounded by an elliptic region in the sense of Mancho et al. (2013) associated with eddy cores and DHTs have to be identified but also the stable and unstable manifolds associated with the latter to find eddy boundaries according to the concept of an eddy in Sect. 2. Those manifolds are visible as singular lines in the color-coded plot of the Lagrangian descriptor $M$ (Fig. 2 (d)-(f), 3 (c)) and the Lagrangian
descriptor $M_V$ (Fig. 2 (a)-(c), 3 (d)) respectively.

How detailed the displayed fine structure of the Lagrangian descriptors $M$ and $M_V$ is represented depends on the chosen value of the time interval $\tau$. It ranges from no clear structure for small $\tau$ (Fig. 2 (a) and (d)) to a detailed structure for large $\tau$ (Fig. 2 (c) and (f)).

From these properties, distinction between DHTs and eddy cores and identification of manifolds, we can conclude that the

Lagrangian descriptor $M_V$ is a suitable method for an automated search for eddies in oceanographic flows. Out of the four considered quantities $M_V$ allows for a clear identification of eddy cores and the stable and unstable manifolds of DHTs necessary to get more insight into the size of eddies with the least number of check criteria. For this reason we suggest to use $M_V$ as the basis for an eddy tracking tool. How these properties of $M_V$ are implemented into an eddy tracking tool is explained for the eddy core in Sect. 4.1 and the eddy size and shape in Sect. 4.3.

## 4  The Lagrangian descriptor $M_V$ as eddy tracking tool

The mean oceanic flow is superimposed by many eddies of different sizes which emerge at some time instant, persist for some time interval and disappear. Consequently, an eddy tracking tool has to detect them at the instance of emergence, track them over their lifetime and detect their disappearance. To classify the different eddies some information about their size is needed too. This way one can finally obtain the time evolution of a size distribution function of eddies.

In this section we apply the modulus of vorticity based Lagrangian descriptor $M_V$ to the hydrodynamic model of a vortex street to test its performance as an eddy tracking tool. We use the local maxima of $M_V$ for an automated search for eddy cores and in addition, the area enclosed by the singular lines of $M_V$ associated with the manifolds of the DHTs as measure for the size of the eddies.

### 4.1  Eddy birth and lifetime

First we check how well $M_V$ detects the birth of an eddy and its lifetime and compare the results to the oceanographic eddy tracking tool box (ETTB) by Nencioli et al. (2010), as well as Eulerian quantities like the Okubo-Weiss parameter and the modulus of vorticity.

The idea of the tracking inspired by Nencioli et al. (2010) is to search for local maxima ($M_V$ and modulus of vorticity) or local minima (Okubo-Weiss, velocity based method by Nencioli et al. (2010)) surrounded by a region of gradient towards the maximum or minimum in a given search window. The size of the search window determines which maximal eddy size can be detected. The eddy is tracked from one time step to the next by searching for an eddy core with the same direction of rotation within a given distance. The choice of this distance depends on the velocity field. It should be in the range of the maximal distance a particle could travel in the timespan of interest. The position of an eddy is logged in a track-list for each eddy at each time step. A track-list that is shorter than a given threshold number of time steps is deleted to focus on eddies which exist longer than this minimum time interval. A detailed description of the algorithms can be found in the supplemental material to this article.

In order to check the accuracy of the eddy tracking algorithm, we use the dimensionless model of the vortex street presented in Sect. 3, since the time instant of birth of the eddies and their lifetimes are given analytically. We measure both quantities for different dimensionless lifetimes $T_c$ and dimensionless vortex strengths of 200, 100 and 50 for the eddy that arises at time $T_c/2$. The rationale behind varying the vortex strength is to estimate how weak an eddy could be to be still reliably detected by the methods. For $M_V$, $\tau$ was chosen as $0.15 \cdot T_c$. The results are presented in Figs. 4 and 5.

In all cases independent of the vortex strength, the results obtained with $M_V$ are close to the analytical $T_c$ (Fig. 4) or the analytical time instant of birth (Fig. 5). All other methods underestimate $T_c$ and overestimate the time instant of birth. Especially in case of ETTB the estimated times depend heavily on the vortex strength. For that method it becomes more and more difficult to detect the eddy as its rotation speed decreases. The reason for the good estimates provided by $M_V$ lies in its construction
which makes use of the history of the eddy (past and future). Hence it can detect eddies earlier than they arise by taking into account the future or detect them longer than they actually exist by looking into the past. $M_V$ is not restricted to the information about the velocity field at one instant of time like the other methods. However, the performance of $M_V$ depends on the chosen value of $\tau$ (Fig. 6). If $\tau$ gets too large in relation to $T_c$, the estimate of the lifetime deviates from the analytical one because the trajectories contain too much of the history of the eddy. There exists a small range of optimal $\tau$ for a certain class of eddies. In
our case the range is between about 15 % and 18 % of the eddy lifetime. We have chosen 15 % of the eddy lifetime, because larger $\tau$ values increase the computational costs for $M_V$, too. The range of the optimal $\tau$ depends crucially on the application. Other applications might need a larger or smaller $\tau$ or a $\tau$ that is a compromise between structures with very different lifetimes. It is also advisable to vary $\tau$ to detect different size and lifetime spectra of eddies.

### 4.2  Robustness of the lifetime detection with respect to noise

Velocity fields describing ocean flows have either a finite resolution when obtained by simulations or contain measurement noise when retrieved from observational data. For this reason, an eddy tracking method has to be robust with respect to fluctuations of the velocity field. Therefore, we explore how the detected eddy lifetime depends on noise added to the velocity data.

To test the influence of noise in a manageable test setup where we know all parameters like e.g. eddy lifetime (here $T_c = 1$)
or vortex strength (here $w = 200$) we use the velocity components $u(x, y, t)$ and $v(x, y, t)$ of the vortex street mentioned in Sect. 3 and add three different types of noise to it mimicking measurement noise that can arise in observations. The result are noisy velocity components $u_N(x, y, t)$ and $v_N(x, y, t)$ for which we calculate Okubo-Weiss, modulus of vorticity and $M_V$ and then apply the different eddy tracking methods. The noise is realised as white Gaussian noise in form of a matrix of normally distributed random numbers of the grid size for each time step multiplied by a factor that is referred to as noise level or noise
strength. The noise level is given dimensionless, because the noise is applied to the dimensionless model of the vortex street presented in Sect. 3.

The different noise types and their motivation are given in the following:

1. type 1: We add white Gaussian noise $\xi(x, y, t)$ of different noise strength $\sigma$ between 0.05 and 0.95 to the velocity components $u(x, y, t)$ and $v(x, y, t)$ of the vortex street. The noise is uncorrelated in space and time. The resulting
velocity components $u_N(x, y, t) = u(x, y, t) + \sigma \cdot \xi_u(x, y, t)$ and $v_N(x, y, t) = v(x, y, t) + \sigma \cdot \xi_v(x, y, t)$ in this case are still periodic but noisy. This type of noise mimics the effect of computing derivatives of observed velocity fields (e.g. by satellites or HF-radar).

2. type 2: We add white Gaussian noise $\xi(x,y,t)$ of different noise strength $\sigma$ between $0.05$ and $0.95$ to the velocity components $u(x,y,t)$ and $v(x,y,t)$ of the vortex street but take into account that the actual noise depends on the velocity itself by taking the maximum of it over the whole spatial grid. The motivation is that the strength of noise depends on the "Signal to noise" ratio. If we have a strong current, it is easy to detect this by a satellite, since the signal-strength is high. This is opposite for slow currents, where the noise level is much higher. Thus, we add white noise that is inversely proportional to the current speed. The noisy velocity components are given as $u_N(x,y,t) = u(x,y,t)+\sigma\cdot\xi_u(x,y,t)/(1+\max\limits_{x,y}(u(x,y,t)))$ and $v_N(x,y,t) = v(x,y,t) + \sigma \cdot \xi_v(x,y,t)/(1+\max\limits_{x,y}(v(x,y,t)))$.

3. type 3: We add white Gaussian noise $\xi(t)$ of different noise strength $\sigma$ between $0.05$ and $0.5$ to the $y$-component of the eddy centres' movement. The equations of the unperturbed velocity field contains a part that describes the movement of the eddy centres (see supplemental material). The motion of the $y$-components of the eddy centres in the unperturbed velocity field $(u, v)$ is given by $y_1(t) = y_0 = -y_2(t)$ where the index 1or 2 refers to the two eddies. The movement of the eddy centres in case of noise is given by $y_{1N}(t) = y_0 + \sigma \cdot \xi(t)$ and $y_{2N}(t) = -y_0 + \sigma \cdot \xi(t)$. This type of noise can be observed if the velocity fields have to rely on georeferencing. For instance, satellite generated velocity fields have to be mapped on a longitude/latitude grid, since the satellite is moving. During this postprocessing step a shift in the georeference is possible, leading to translational shifts and thus to type 3 noise. However, a high noise level of type 3 is not very likely. If one deals with typical geophysical applications, which have a grid resolution of the order 1 to 10 km, the georeferencing errors are mostly small compared to the grid cell size. For this reason, the considered noise levels for type 3 noise are smaller than for type 1 and 2.

To explore the impact of noise systematically, we have used different noise strengths. For each noise strength $\sigma$ 1000 real-izations of the white Gaussian noise were calculated. In the resulting velocity fields we estimated the lifetime of each eddy that undergoes a whole life-cycle within the simulation time. The plotted eddy lifetimes obtained with all different tracking methods are medians of the distributions of the lifetimes for the 1000 realizations per noise strength (Fig. 7, 8, 9).

The three types of noise illustrate different advantages and disadvantages of $M_V$ compared to the other methods. In case of type 1 noise, $M_V$ gives the best estimate of the lifetime compared to all other methods independent of the increasing noise level. The reason why the error of the estimate in case of $M_V$ does not increase with increasing noise level is that $M_V$ is a measure that is based on an integral. Integrating over accumulated uncorrelated noise along the trajectory from past to future can be considered as a smoothing process. Also the ETTB by Nencioli et al. (2010) gives a good result independent of the increasing noise level, because the signal to noise ratio is small. The minimum of the velocity that is the key-signal for determining the eddy core in their method remains a local minimum in the contour plot of the velocity. However, with increasing noise level we find an increase of outliers for the ETTB by Nencioli et al. (2010) and $M_V$ (boxplot not shown here). The performance of modulus of vorticity and the Okubo-Weiss parameter decreases as expected with increasing noise level while the distribution increases in width (Fig. 7). The reason is that the noise gets so large that it increasingly disturbs the key-signal for an eddy core until no distinct eddy core can be identified anymore.

In case of type 2 noise, $M_V$ and ETTB show a similar behaviour as in case of type 1 noise. Both yield good results independent

of the noise level. This is again due to the smoothing process in case of $M_V$. The modulus of vorticity performs even better than $M_V$ in case of small noise levels, but its performance drops below the results of $M_V$ with increasing noise level (Fig. 8). The reason is that the key-signal for determining an eddy core using the modulus vorticity is stronger in case of small noise levels and gets disturbed by the noise with increasing noise level. As expected, the performance of Okubo-Weiss decreases

with increasing noise level. In contrast to type 1 noise, Okubo-Weiss can identify eddy cores even in case of strong noise, because the key-signal for an eddy core is less disturbed.

In case of type 3 noise, $M_V$ yields an estimate of the lifetime with the largest error (Fig. 9). In this case noisy trajectories that start close to each other diverge fast, while the ones with no noise have a similar dynamical evolution. This divergence due to noise leads to a loss of structure in space that can be interpreted as a weakening of the correlation between neighbouring

trajectories. This effect is strongest in case of $M_V$ because it integrates over time and so neighbouring trajectories that have similar values of $M_V$ in case of no noise yield very different values of $M_V$ due to the divergence of the trajectories. As a consequence no clear structure in $M_V$ can be identified. This effect increases with the noise level.

Also for the other methods noise of type 3 affects strongly the identification of the eddy core because the weakening of the correlation between neighbouring points disturbs the key-signal of an eddy core (a local minimum or maximum in a certain

domain). The error in estimating the lifetime increases with increasing noise level. In all cases the number of outliers in the boxplot (not shown here) increases with the noise level.

As a consequence, none of the methods performs in an optimal way when the noise displaces the eddy cores during their motion. This disadvantage will lead to deviations in the lifetime statistics for eddy tracking based on observational data. However, the error in georeferencing of satellite images (which is mimicked by type 3 noise) is mostly small. For special applications,

a georeferencing error of smaller than 1/50 pixel is achievable (Leprince et al. (2007)). Eugenio and Marqués (2003) show that with reasonable effort a mapping error smaller than 0.5 pixel is possible, if fixed landmarks (coastlines, islands) are on the images. With the increase in earth orbiting satellites and thus the increase in available images, it can be assumed that this error will drop even more (Morrow and Le Traon (2012)). If numerically generated velocity fields are used, noise of type 3 is completely absent. Here the evolution of neighbouring trajectories is smooth and correlated.

In summary, $M_V$ can be used for the detection of eddies and the estimate of eddy lifetimes for velocity fields with and without noise and yields good results independent of the noise level in case of type 1 and 2 noise. However, one has to take into account that the velocity field should not be too noisy and that one has to chose a $\tau$ that fits the problem. The Lagrangian descriptor $M_V$ has an additional advantage in detecting arising eddies earlier than other methods due to collecting information along the trajectory from past to future. This can be useful in the identification of regions that will be eddy dominated in the further

evolution of the flow.

### 4.3   Detecting eddy sizes and shapes

Beside its lifetime an eddy is characterized by its size. In the following we will estimate the eddy size and shape using the the Lagrangian descriptor $M_V$ based on the modulus of vorticity and compare the results to the size detected by the ETTB by Nencioli et al. (2010). In this way, we demonstrate the differences between the Eulerian and Lagrangian point of view on the

eddy size and shape.

As mentioned in Sect. 2 the estimation of the eddy shape and size from the Lagrangian point of view is based on the idea that the boundaries of the eddy are linked to manifolds of DHTs that surround the eddy (Branicki et al. (2011); Bettencourt et al. (2012)). These manifolds cannot be crossed by any trajectories and, therefore, trajectories starting inside the manifolds are trapped in the eddy. Defining the boundaries in this way one can estimate the trapping region or volume that is transported by an eddy.

The Lagrangian descriptor $M_V$ displays singular lines that correspond to manifolds. Therefore, the shape detection algorithm searches for the largest closed contour line of $M_V$ for which $M_V$ is an extremum and which surrounds an eddy core found with $M_V$. This contour line, extracted from $M_V$ with the MATLAB function contourc and along which the gradient of $M_V$ is large, should be a line on or very close to a singular line displayed by $M_V$ corresponding to a manifold and will give an estimate of the eddy boundary.

The ETTB by Nencioli et al. (2010) gives an Eulerian view on the eddy shape by defining the eddy boundaries as the largest closed streamline of the streamfunction around the eddy centre where the velocity still increases radially from the centre. The contour lines as well as the streamlines are extracted in a given search window which is centred on the eddy core.

The comparison of the different views on the eddy size and shape is presented in Fig. 10 for the vortex street without (a) and with noise of type 1, 2 and 3 ((b)-(d)). The size detected with the ETTB by Nencioli et al. (2010) is much smaller than the size based on the Lagrangian view (Fig. 10 (a)-(c)). Additionally, the evolution of the eddy is captured by both methods even in case of strong type 1 and 2 noise (Fig. 10 (b) and (c)). Here, the eddy boundaries in case of noise show small irregularities due to the noise. In general, the eddy boundary computed based on $M_V$ is detected earlier and shows more growing and shrinking during the evolution of the eddy than the eddy boundary extracted by the ETTB. This is due to the conceptual idea of $M_V$ that contains the history of the trajectories. As shown in Sect. 4.2 this leads to problems in case of a velocity field with type 3 noise (although significant type 3 noise levels are very unlikely). If the noise level is too large, no structure neither a clear eddy core nor a clear eddy boundary can be detected (Fig. 10 (d)) within $M_V$. But if an eddy core can be detected as in case of the left eddy in Fig. 10 (d) the eddy shape detection based on $M_V$ gives an idea of the size and the noisy eddy boundary.

In a real oceanic flow eddies of different lifetime, size and shape will occur simultaneously. As an outlook, how different eddy shapes and sizes can be detected in real oceanic flow fields, we apply our approach to a velocity field of the western Baltic Sea for May 2009. The Baltic Sea is a good testbed, since the tides there are negligible and the entire eddy dynamics is driven by baroclinic instabilities, frontal dynamics and the interaction with topography. An extended eddy statistics in the central Baltic Sea based on $M_V$ will be the content of further research.

A triple-nested circulation model was used to simulate the flow fields in the western Baltic Sea. The innermost model domain was discretised in the horizontal with a spatial resolution of 1/3 nautical mile (approx. 600 m). The model domain covers the Danish Straits and the western Baltic. The open boundaries are located in the Kattegat and at the eastern rim of the Bornholm Basin. In the vertical 50 terrain-following adaptive layers, with a zooming toward stratification were used. The setup is identical to the one used by Klingbeil et al. (2014) or Gräwe et al. (2015b). There, a detailed description and validation of the present setup can be found. At the open boundaries of the model domain, the water elevations, depth averaged currents, as well as

salinity and temperature profiles are prescribed. This external forcing was taken from a model of the North Sea-Baltic Sea with a horizontal resolution of 1 nautical mile and 50 vertical layers. To account for large scale variations and remotely generated storm surges, the North Sea-Baltic Sea model was nested into a depth-averaged storm surge model of the North Atlantic with a resolution of 5 nautical miles. The atmospheric forcing was derived from the operational model of the German Weather Service

with a spatial resolution of 7 km and temporal resolution of 3 hours. A more detailed description of the model system is given by Gräwe et al. (2015a). The flow fields for May 2009 were taken out of a running simulation covering the period 1948-2015. The velocity field was interpolated to an equidistant spacing of 1 m and finally averaged over the upper 10 m to produce a "quasi" two-dimensional field. The temporal resolution was set to one hour to resolve for instance inertial oscillations.

We have calculated $M_V$ for 11 May 2009 1:00 am with $\tau = 36$ h and applied the eddy tracking based on $M_V$. A $\tau$ value of 36 h

corresponds to 15 % of an eddy lifetime of approx. 10-12 days, which was reported previously by Fennel (2001). In contrast to the test case of the vortex street, we do not expect that the eddies are perfectly circular. To account for deformed and distorted eddies, we had to introduce a threshold for the convexity deficiency to eliminate contours that are only made out of filaments and are no eddy in the sense of oceanography. We set the threshold to 11% difference between the area of the convex hull of the points that form the boundary and the area enclosed by the boundary itself normalised to the area enclosed of the boundary.

This definition of convexity deficiency is according to Haller et al. (2016). Please note, that we still allow to detect contours that cover eddy merging and decay processes, which are characterized by filaments.

Fig. 11 shows the eddy boundaries detected with the method based on $M_V$ (red) and the ETTB by Nencioli et al. (2010) (black) at 11 May 2009 1:00 am for the same search window size. There are several differences between the number and shapes of eddies which has to be explained. 150 eddies can be detected with the method based on $M_V$, whereas the ETTB detects only

24 eddies at the same instant of time. One reason for the differences is that $M_V$ contains the information of the velocity field of a time interval, namely 11 May 2009 1:00 am $\pm 36$ h. Each eddy that exists, starts to arise, merges with another eddy or dies within this time interval leaves a footprint in $M_V$ like the many of small eddies visible in $M_V$. How strong this footprint is visible in $M_V$ depends on the choice of $\tau$. Therefore, the number of eddies detected with the method based on $M_V$ has to be compared with the number of eddies detected with the ETTB by Nencioli et al. (2010) in the whole time interval that is

covered by $M_V$.

The black dots in Fig. 11 are the eddy cores detected with the ETTB by Nencioli et al. (2010) within the time interval 11 May 2009 1:00 am $\pm 36$ h. In total, 339 eddies are detected which exist between less than one hour and 72 hours. For some eddies we will discuss exemplarily why they are detected by one of the methods and not by the other to illustrate which different problems have to be taken into account if one interprets the results of the different methods.

Close to or within the eddy 1, 2 and 3 detected by the tracking based on $M_V$ are several eddy cores detected by the ETTB by Nencioli et al. (2010) if one takes into account the whole time interval. At 11 May 2009 1:00 am the ETTB does not detect eddy 1, 2 and 3 because they are to weak or do not exist yet. By contrast, the eddy detection method based on $M_V$ detects them due to the construction of $M_V$ as an integral over time. For eddy 4 only a few eddy cores are detected by the ETTB by Nencioli et al. (2010) for the whole time interval, probably the eddy is to weak and lives to short to be seen as a structure in

$M_V$. In case of eddy 5 the method based on $M_V$ does not detect an eddy although the ETTB by Nencioli et al. (2010) detects

several eddy cores in the region. One reason could be that the eddy arises, moves a lot and dies within the time interval such that $M_V$ only captures a blurred structure of the eddy that does not fulfil the convexity criterion. Eddy 6 is not detected by the method based on $M_V$, although the eddy boundary is obvious in the structure of $M_V$. The reason is that the choice of the search window size for the eddy core detection determines if an eddy core is detected or not. An enlarged search window could solve this problem for eddy 6, but a larger search window influences the number of detected eddies. A solution could be an eddy core search independent of the search window size.

A general problem, which arises when using surface velocity fields, is that this velocity field is not divergence free. Although, we have checked that the vertical velocity is small compared to the horizontal ones, there is still a finite residual left. However, we still assume that the velocities are two dimensional. Applying the ETTB by Nencioli et al. (2010) to these "quasi" 2d fields does not cause difficulties, since the algorithm works on an instantaneous snapshot - a frozen velocity field. Thus, the error made by the 2d assumption is small. The situation changes when employing a Lagrangian descriptor. During the integration interval $[t^* - \tau \; t^* + \tau]$, $M_V$ accumulates these residuals. Therefore, $M_V$ can show structures that seems to be eddies but are regions of a stronger vertical velocity or Lagrangian divergence (Jacobs et al. (2016)). Therefore, the number of eddies of both methods will include false positives.

In summary, the method based on the Lagrangian descriptor $M_V$ can be used for the detection of eddy boundaries that are acting as boundaries of a trapping region. Comparing the latter to boundaries detected with the ETTB by Nencioli et al. (2010) leads to large differences in the shape and in the size. Those deviations are due to the difference in the definition of the boundary and yields in case of the vortex street much smaller sizes of the eddies in case of the ETTB by Nencioli et al. (2010). Another advantage of the method based on the Lagrangian descriptor $M_V$ is that it even shows filament structures of the eddy boundary in contrast to the ETTB by Nencioli et al. (2010) visible in the example of the western Baltic Sea. These filaments can be linked to the dynamics of the eddy, e.g. as it starts interacting, merging, or repelling with other eddies or fading out. Though these filamental shapes of eddies might not be eddies according to a more strict mathematical definition of an eddy boundary as in Branicki et al. (2011); Haller et al. (2016), but they are still important structures in the flow from an oceanographic point of view and should be considered in a census of eddies.

Nevertheless, one has to take into account that the detection of eddy shapes by the method based on the Lagrangian descriptor $M_V$ is restricted by the choice of $\tau$. In highly dynamical velocity fields like the example of the Baltic Sea not all structures can be resolved by the same $\tau$ which leads to a compromise for $\tau$. This choice of $\tau$ influences if an eddy can be detected by the method based on $M_V$ and not by the ETTB by Nencioli et al. (2010) or the other way round.

The method to detect shapes should be chosen based on the question which type of shapes one is interested in and the results of the method should be handled with care.

## 5   Discussion and conclusion

We have shown, that the Lagrangian descriptor $M_V$ based on the modulus of vorticity provides good insights into the structure of a hydrodynamic flow. It can be used to identify eddy cores as well as distinguished hyperbolic trajectories. Eddy cores can

be found as local maxima of $M_V$, while DHTs correspond to minima of $M_V$. Hence, compared to the Lagrangian descriptor $M$ based on the arc length, it does not need an additional criterion to distinguish between eddy cores and DHTs. As any other Lagrangian descriptor it displays singular lines that can be linked to the stable and unstable manifolds of the DHTs which allows for a simultaneous estimate of the boundaries of the eddies to get an assessment of their size and shape. These features make the quantity $M_V$ suitable for designing an eddy tracking tool, which should be able to detect eddy cores, to track them over time, and additionally to provide information about the eddies' lifetime, size and shape. Moreover, the eddy tracking should be robust with respect to velocity fields corroborated with errors in case the velocity field is extracted from observations.

To test all those properties in practice we have first used some velocity fields, which are constructed in such a way that the lifetimes of eddies are given analytically. It turns out, that the Lagrangian descriptor $M_V$ is superior in estimating lifetimes compared to the other considered methods. This is due to its definition as an integral which takes the history into account. Eulerian methods like Okubo-Weiss or the ETTB by Nencioli et al. (2010) detect eddies too late and underestimate their lifetime. The formulation of $M_V$ as an integral is also beneficial in case of different types of noise. However, none of the tested methods can deal in a convincing way with type 3 noise which mimics errors to shifts in georeferencing.

A general problem of any Lagrangian descriptor including $M$ and $M_V$ is that the resolution of the structures to be detected depends on the chosen time $\tau$. Structures that live too short in relation to the chosen $\tau$ cannot be resolved and will be missed. Hence the choice of $\tau$ contains a decision which time scale and subsequently which eddy lifetime will be resolved.

The example of the velocity field of the western Baltic Sea shows that eddy tracking based on $M_V$ is able to detect the essential eddies that are visible in the velocity field and also detected by the ETTB by Nencioli et al. (2010). Furthermore, it detects eddies that cannot be detected by the ETTB at this instant of time $t^*$, but was or will be detected by the ETTB at a earlier or later instant of time within the time interval $[t^* - \tau \ t^* + \tau]$. Nevertheless, one has to be aware that both ETTB and the eddy tracking based on $M_V$ give false positives. The reason could be that structures of strong vertical velocity are identified as eddies. On the other hand false negatives can arise if (i) the eddies are too weak or (ii) the chosen $\tau$ value is too large or too small or (iii) the search window is too large or too small.

In general, the choice of the detection method depends on the questions asked. If one is only interested in tracking eddy cores Eulerian methods are a good choice. By contrast, Lagrangian methods gives a more detailed view on the dynamics and provide a more physical estimate of the eddy size. Especially this feature, which describes the fluid volume trapped in an eddy promises to be more useful for applications that consider the growth of plankton populations in oceanic flows. For the latter it has been shown that eddies can act as incubators for plankton blooms due to the confinement of plankton inside the eddy (Oschlies and Garçon (1999); Martin (2003); Sandulescu et al. (2007)).

*Author contributions.* Rahel Vortmeyer-Kley developed the idea of the eddy tracking tool based on the Lagrangian descriptor $M_V$ and implemented it. Ulf Gräwe supervised the oceanic questions of this work and provided the velocity field for the western Baltic Sea. The overall supervision was done by Ulrike Feudel. All authors contributed in preparing this manuscript.

*Acknowledgements.* Rahel Vortmeyer-Kley would like to thank the Studienstiftung des Deutschen Volkes for a doctoral fellowship. The financing of further developments of the Leibniz Institute of Baltic Sea Research monitoring program and adaptations of numerical models (STB-MODAT) by the federal state government of Mecklenburg-Vorpommern is greatly acknowledged by Ulf Gräwe.

The authors would like to thank Jan Freund, Ana Mancho, Matthias Schröder, Wenbo Tang, Tamás Tél and Alfred Ziegler for stimulating discussions.

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

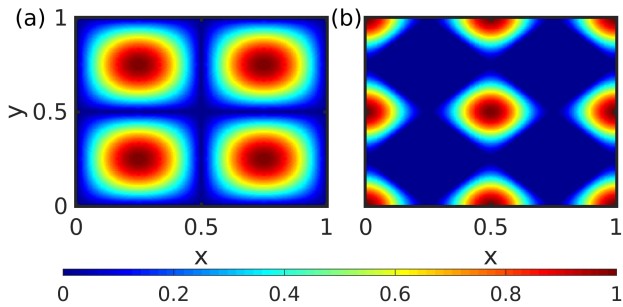

**Figure 1.** Colour-coded representation of the modulus of vorticity (a), Okubo-Weiss-Parameter (b) for the eddy field in Eq. (6). All plots are normalized to the maximum value.

Sandulescu, M., Hernández-García, E., López, C., and Feudel, U.: Kinematic studies of transport across an island wake, with application to Canary islands, Tellus A, 58, 605–615, 2006.

Sandulescu, M., López, C., Hernández-García, E., and Feudel, U.: Plankton blooms in vortices: the role of biological and hydrodynamic timescales, Nonlinear Proc Geoph, 14, 443–454, 2007.

5 Sturman, R. and Wiggins, S.: Eulerian indiators for predicting and optimazing mmixing quality, New J Phys, 11, 075 031, 2009.

Tang, W. and Luna, C.: Dependence of advection-diffusion-reaction on flow coherent structures, Phys Fluids, 25, 106 602–1–19, 2013.

Thacker, W. C., Lee, S.-K., and Halliwell, G. R.: Assimilating 20 years of Atlantic XBT data into HYCOM: a first look, Ocean Model, 7, 183–210, 2004.

Weiss, J.: The dynamics of enstrophy transfer in two-dimensional hydrodynamics, Physica D: Nonlinear Phenomena, 48, 273–294, 1991.

10 Wiggins, S.: The dynamical systems approach to Lagrangian transprt in oceanic flows, Annu Rev Fluid Mech, 37, 295–328, 2005.

Wiggins, S. and Mancho, A.: Barriers to transport in aperiodically time-dependent two-dimensional velocity fields: Nekhoroshev's theorem and "Nearly Invariant" tori, Nonlinear Proc Geoph, 21, 165–185, 2014.

Wilson, M. M., Peng, J., Dabiri, J. O., and Eldredge, J. D.: Lagrangian coherent structures in low Reynolds number swimming, J Phys-Condens Mat, 21, 204 105, 2009.

15 Wischgoll, T. and Scheuermann, G.: Detection and visualization of closed streamlines in planar flows, IEEE T Vis Compu Gr, 7, 165–172, 2001.

Yang, Q., Parvin, B., and Mariano, A.: Detection of vortices and saddle points in SST data, Geophys Res Lett, 28, 331–334, 2001.

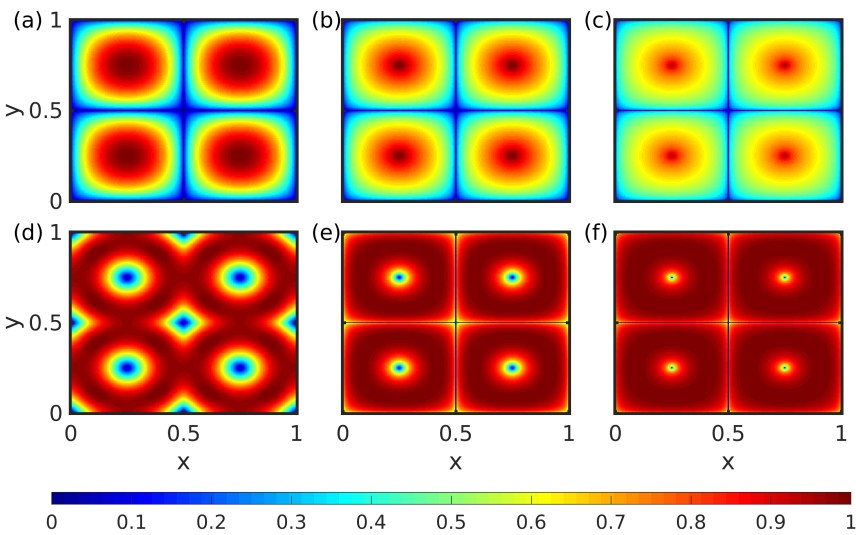

**Figure 2.** Colour-coded representation of the Lagrangian descriptor $M_V$ (a)-(c) and the Lagrangian descriptor $M$ (d)-(f) for the eddy field in Eq. (6) with $\tau$ chosen as 0.5 ((a) and (d)), 25 ((b) and (e)) and 100 ((c) and (f)). All plots are normalized to the maximum value.

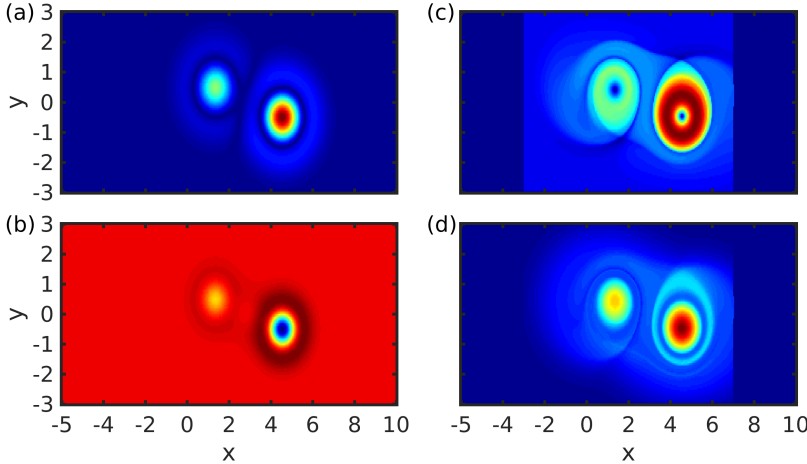

**Figure 3.** Modulus of vorticity (a), Okubo-Weiss parameter (b), Lagrangian descriptor $M$ (c) and Lagrangian descriptor $M_V$ (d) for the hydrodynamic model of a vortex street at $t = 0.151$ with $\tau = 0.15$, normalized to the maximum value. Blue colours indicate small values yellow large values of the depicted quantity. The dark blue regions in (c) and (d) are regions where the trajectories have left the region of interest.

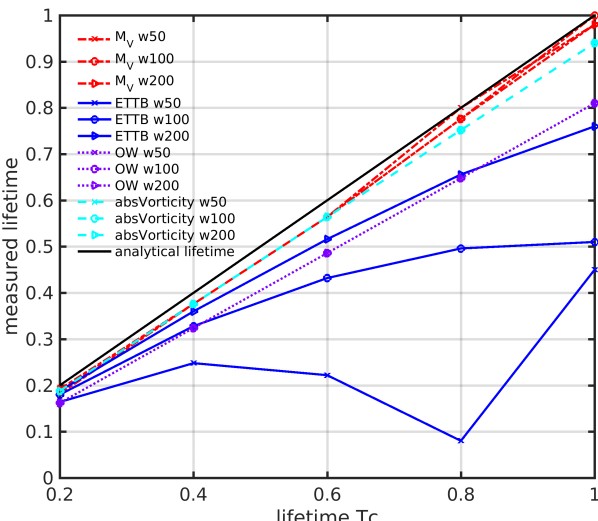

**Figure 4.** Eddy lifetime estimated with Okubo-Weiss (OW, violet), modulus of vorticity (absVorticity,cyan), $M_V$ (red) and the eddy tracking tool box (ETTB, blue) by Nencioli et al. (2010) for vortex strength $w$ 50, 100 and 200. The black diagonal depicts the analytical lifetime.

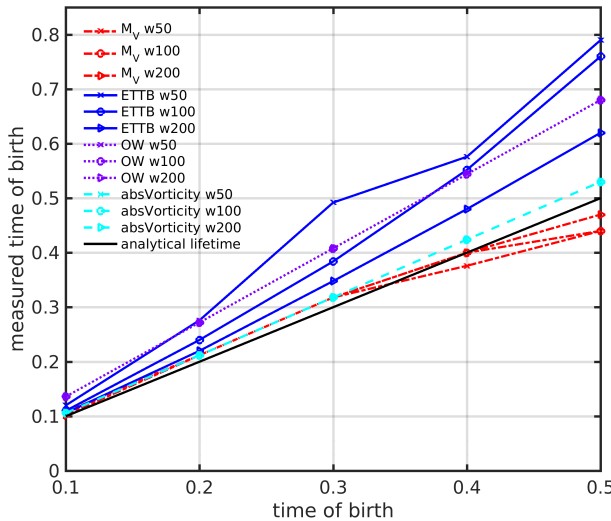

**Figure 5.** Time of birth of an eddy estimated with Okubo-Weiss (OW, violet), modulus of vorticity (absVorticity, cyan), $M_V$ (red) and the eddy tracking tool box (ETTB, blue) by Nencioli et al. (2010) for vortex strength $w$ 50, 100 and 200. The black diagonal depicts the analytical time of birth of an eddy.

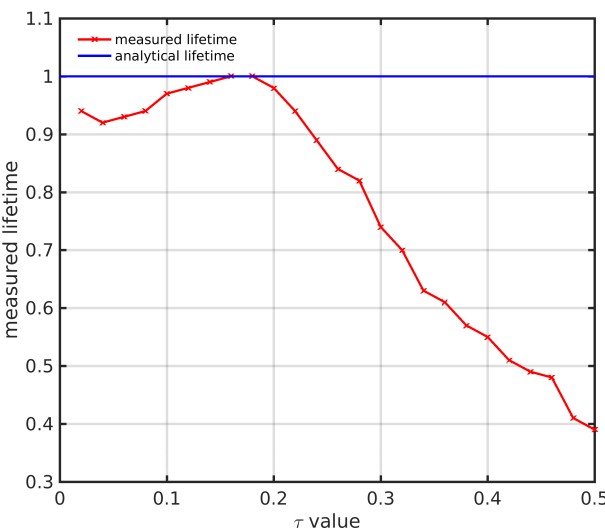

**Figure 6.** Measured lifetime of an eddy obtained by means of $M_V$ (red line) versus the chosen $\tau$ (analytical lifetime $T_c = 1$ (blue line), vortex strength 200).

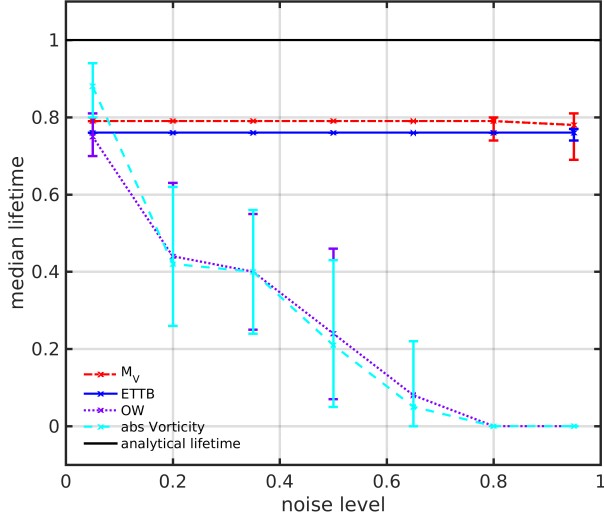

**Figure 7.** Measured median lifetime obtained by different methods (Okubo-Weiss (OW, violet), modulus of vorticity (absVorticity, cyan), $M_V$ (red) and the eddy tracking tool box (ETTB, blue) by Nencioli et al. (2010)) depending on the noise level. The computations have been performed in a velocity field mimicking a vortex street with added white Gaussian noise (type 1 noise with 1000 noise realizations). The errorbars indicate the whiskers of the distribution in the boxplot (not shown here) corresponding to approximately $\pm 2.7\sigma$.

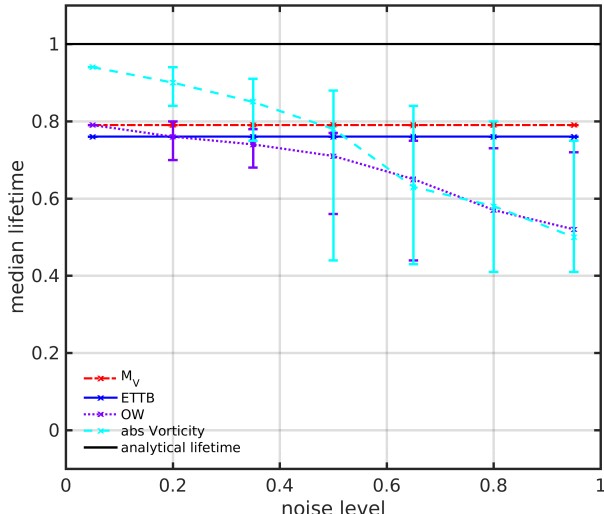

**Figure 8.** Measured median lifetime obtained by different methods (Okubo-Weiss (OW, violet), modulus of vorticity (absVorticity, cyan), $M_V$ (red) and the eddy tracking tool box (ETTB, blue) by Nencioli et al. (2010)) depending on the noise level. The computations have been performed in a velocity field mimicking a vortex street with type 2 noise (1000 noise realizations). The errorbars indicate the whiskers of the distribution in the boxplot (not shown here) corresponding to approximately $\pm 2.7\sigma$.

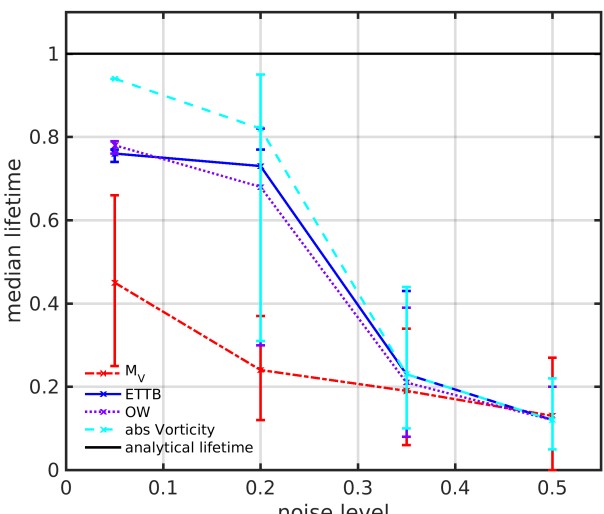

**Figure 9.** Measured median lifetime obtained by different methods (Okubo-Weiss (OW, violet), modulus of vorticity (absVorticity, cyan), $M_V$ (red) and the eddy tracking tool box (ETTB, blue) by Nencioli et al. (2010)) depending on the noise level. The computations have been performed in a velocity field mimicking a vortex street with type 3 noise (1000 noise realizations). The errorbars indicate the whiskers of the distribution in the boxplot (not shown here) corresponding to approximately $\pm 2.7\sigma$.

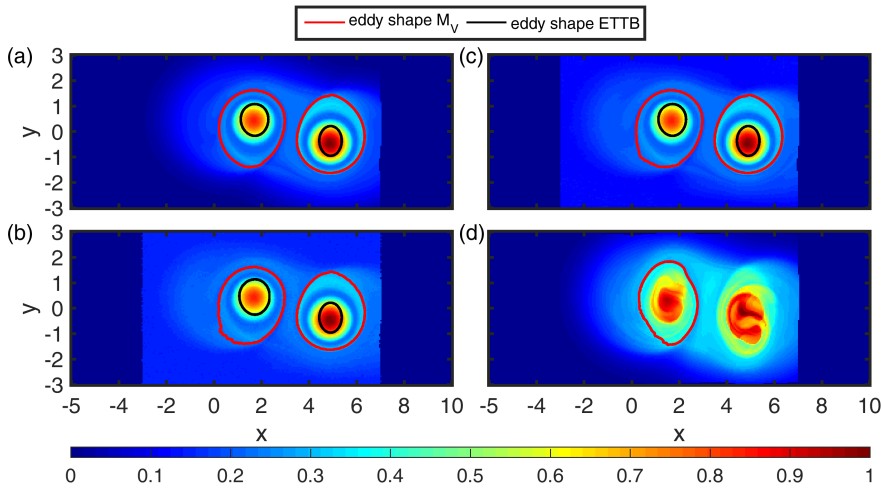

**Figure 10.** Eddy boundaries detected with the method based on $M_V$ (red line) and with the eddy tracking tool by Nencioli et al. (2010) (black line) at $t = 0.201$. (a) $M_V$ without noise, (b) $M_V$ with type 1 noise of noise level 0.95, (c) $M_V$ with type 2 noise of noise level 0.95, (d) $M_V$ with type 3 noise of noise level 0.5. The $\tau$ value is chosen as 0.15 $T_c$ with $T_c = 1$. The dark blue regions are regions where the trajectories have left the region of interest. All plots are normalized to the maximum value.

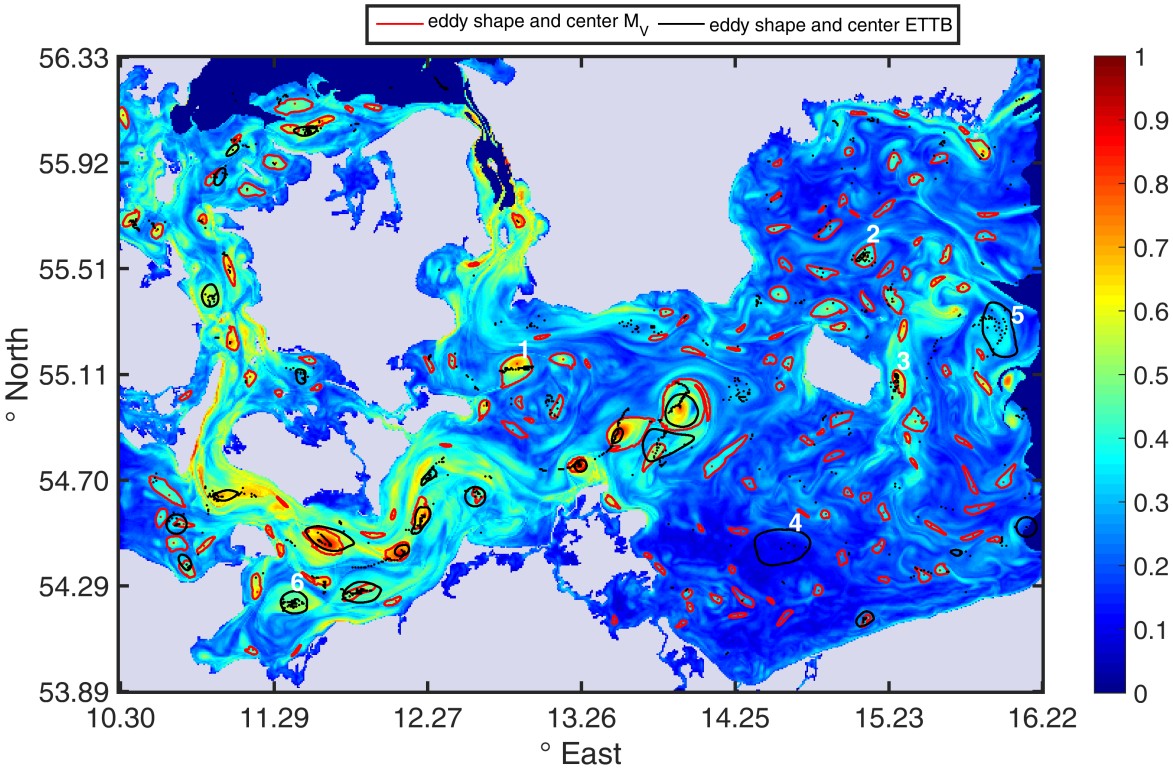

**Figure 11.** $M_V$ for the western Baltic Sea for 11 May 2009 1:00 am with $\tau = 36$ h normalized to the maximum value of $M_V$. The red lines are the eddy boundaries and red dots the eddy cores detected with the method based on $M_V$ respectively. The black lines are the eddy boundaries detected with the ETTB by Nencioli et al. (2010) at 11 May 2009 1:00 am. The black dots are the eddy cores detected with the ETTB by Nencioli et al. (2010) within the time interval 11 May 2009 1:00 am $\pm 36$ h. The dark blue regions are areas where the trajectories have left the domain of interest, light grey regions indicate land.