# Peer review of "Detecting and tracking eddies in oceanic flow fields: A Lagrangian descriptor based on the modulus of vorticity"

_Nonlinear Processes in Geophysics, 2016_

## Short Comment (SC1) · 7 Mar 2016

A theory and computational procedure of identifying material eddies objectively from the Lagrangian-Averaged Vorticity Deviation (LAVD) has already been developed in the following manuscript:

http://arxiv.org/abs/1506.04061

This manuscript has been posted on-line in arxiv.org since June 2015, and it is now in press at the Journal of Fluid Mechanics.

The above JFM manuscript gives:

[Figure]

(1) A precise mathematical definition of vortex cores and vortex boundaries through the notion of rotational coherence, valid both in 2D and 3D

(2) An exact decomposition of the deformation gradient that isolates an objective material rotation angle (dynamic rotation) for the assessment of rotational coherence.

(3) A theorem this objective material rotation angle is precisely equal to the LAVD.

(4) A theorem on how LAVD-based vortex cores will prevail as attractors/repellers for inertial particle motion in geostrophic eddies.

(5) A related instantaneous Eulerian vortex definition that is also objective, using the derivative of the LAVD.

(6) A total of 6 (six) numerical examples, both 2D and 3D, both steady and unsteady, including a 2D unsteady and a 3D unsteady ocean data set.

(7) Illustration in all examples that all LAVD-based material eddies do remain rotationally coherent under advection (no transverse filamentation arises in their boundaries)

(8) Comparison with two other available objective eddy detection methods.

Therefore, a rigorous and frame-independent theory has already been developed for the same objective that is pursued by the Authors of this NPG submission in a heuristic and frame-dependent fashion. A new feature in their manuscript is the lifetime assessment for a larger set of eddies. This can, however, be carried out (now in an objective and mathematically precise fashion) with LAVD, at the same computational cost, with precisely defined eddy boundaries and eddy cores, and with mathematically guaranteed rotational coherence under Lagrangian advection.

---

## Referee Comment (RC1) · Anonymous Referee #1 · 10 Mar 2016

**Referee Report on "Detecting and tracking eddies in oceanic flow fields: A vorticity based Euler-Lagrange method" by R. Vortmeyer-Kley, U. Grawe, and U. Feudel**

Potentially this paper contains some results that may be of interest to NPG, but not in its present form. The main issue is that the authors describe the method of Lagrangian descriptors in a variety of misleading and wrong ways, and they also have neglected to mention a large amount of relevant literature.

The authors develop a new Lagrangian descriptor based on integrating the magnitude of the vorticity (an Eulerian quantity) along trajectories. They claim this is a new "Eulerian-Lagrangian descriptor". However, their approach follows exactly the methodology for Lagrangian descriptors described in the following paper:

> A. M. Mancho, S. Wiggins, J. Curbelo, C. Mendoza Lagrangian descriptors: A method for revealing phase space structures of general time dependent dynamical systems, Communications in Nonlinear Science and Numerical Simulation, 18(12), 3530 3557 (2013)

At the bottom of page 3532 of this paper it is stated that Lagrangian descriptors can be constructed by integrating any "bounded positive intrinsic physical or geometric property of the velocity field...". Certainly the magnitude of the vorticity satisfies that criteria, but also the magnitude of the velocity field, which is a common Lagrangian descriptor used in the abovementioned paper (henceforth, Mancho et al 2013). It just so happens that the integral of the square root of the magnitude of the velocity field along trajectories has the interpretation of arclength, but it is still of the same character as the quantity studied by the authors who wish to rename the quantity "Eulerian-Lagrangian descriptor". This is completely misleading and contrary to the methodology introduced in Mancho et al 2013. Indeed, all of the quantities in Mancho et al. 2013 would then be ''Eulerian-Lagrangian descriptors" in the terminology of the authors of the paper under review as all of the quantities of Table 1 in Mancho et al. 2013 proposed for the construction of Lagrangian descriptors are Eulerian quantities. In this sense *all* Lagrangian descriptors are constructed from Eulerian and Lagrangian quantities, with the purpose of providing Lagrangian transport information. So there is nothing new "methodologically" in this paper.

The authors claim that Lagrangian descriptors are not objective, and justify this claim by referring to reference [15]. However, we have looked at reference [15], and I cannot find any proof of non-objectivity for Lagrangian descriptors in that reference. If the authors are going to make such a strong claim, then they must provide a reference to a proof of their claim. With respect to objectivity, on page 18 the authors state:

*... it is a heuristic method that lacks objectivity. This can be problematic since it might lead to failure in the detection of some eddies.*

I do not believe this. Can the authors provide *one* example where *lack of objectivity* can lead to non-detection of eddies? Which eddies? As it stands, the authors are making an unsupported statement. This statement of the authors illustrates the fact that the notion of ''objectivity'' is a total red herring for this type of analysis. This is discussed in Section 4.3 of C. Mendoza and A. M. Mancho, The Lagrangian description of aperiodic flows: a case study of the Kuroshio Current, Nonlinear Processes in Geophysics 19 (4) (2012) 449-472. Moreover, in the paper by K. Ide, D. Small, and S. Wiggins, Distinguished hyperbolic trajectories in time dependent fluid flows: analytical and computational approach for velocity fields defined as data sets, Nonlinear Processes in Geophysics, 9(3-4), 237 ? 263 (2002) there is a proof in the appendix that hyperbolic trajectories are preserved under time-dependent transformations that grow "sub-exponentially" in time (obviously, if the time dependence of the transformation is "strongly exponential" it can "cancel" the hyperbolicity).

The authors are proposing what they refer to as a new characterization of eddies based on an elliptic region bounded by segments of stable and unstable manifolds of a hyperbolic trajectory. This allows the lobe dynamics mechanism to control transport in and out of the elliptic region. However, a careful development of eddies from this point of view has already been given in the following references:

> M. Branicki and S. Wiggins, Finite-time Lagrangian transport analysis: stable and unstable manifolds of hyperbolic trajectories and finite-time Lyapunov exponents, Nonlin. Processes Geophys., 17, 136 (2010).

> M. Branicki, A. M. Mancho, and S. Wiggins, A Lagrangian description of transport associated with a front-eddy interaction: Application to data from the North-Western Mediterranean Sea, Physica D, 240(3), 282-304, (2011).

Moreover, besides eddies defined in this way using data sets, kinematic models possessing transient eddies are also developed. These have the advantage of allowing detailed study and precise investigations. It is also shown in the Branicki and Wiggins paper that FTLE will often fail to capture the detailed structure of eddies defined in this way.

The authors claim that Lagrangian descriptors cannot detect eddies. I find this to be a very surprising statement based on several papers in the literature that characterize eddies in the Gulf stream and Gulf of Mexico, for example, in terms of Lagrangian descriptors. See, for example:

> C. Mendoza, A. M. Mancho, M.-H. Rio. The turnstile mechanism across the Kuroshio current: analysis of dynamics in altimeter velocity fields. Nonlinear Proc. Geoph 17 (2010), 2, 103-111.

> C. Mendoza, A. M. Mancho. The Lagrangian description of aperiodic flows: a case study of the Kuroshio Current. Nonlinear Processes in Geophysics 19 (4) (2012) 449-472.

C. Mendoza, A. M. Mancho, S. Wiggins. Lagrangian Descriptors and the Assesment of the Predictive Capacity of Oceanic Data Sets. Nonlinear Processes in Geophysics 21 (2014) 677-689.

V. J. Garcia-Garrido, A. M. Mancho, S. Wiggins, C. Mendoza. A dynamical systems approach to the surface search for debris associated with the disappearance of flight MH370. Nonlinear Processes in Geophysics 22 (6) (2015) 701-712.

There has been a literature developed in recent years related to finding Eulerian quantities that allow one to make conclusions about certain Lagrangian transport phenomena.

R. Sturman and S. Wiggins, Eulerian indicators for predicting and optimizing mixing quality, New J. Phys., 11, 075031 (2009).

K. L. McIlhany, D. Mott, E. Oran and S. Wiggins, Optimizing mixing in lid-driven flow designs through predictions from Eulerian indicators. Physics of Fluids, 23(2), 082005 (2011).

K. L. McIlhany and S. Wiggins, Eulerian indicators under continuously varying conditions. Physics of Fluids, 24, 073601 (2012).

K. L. McIlhany, S. Guth, and S. Wiggins, Lagrangian and Eulerian analysis of transport and mixing in the three dimensional, time dependent Hill's spherical vortex, Physics of Fluids, 27, 063603 (2015).

On page 2 the authors state that:

*Another more heuristic approach is the computation of distinguished hyperbolic trajectories (DHT) and their stable and unstable manifolds to identify Lagrangian coherent structures in a flow.*

The word "heuristic" and the phrase "to identify Lagrangian coherent structures in a flow" are used very bizarrely, and incorrectly, here. First, there is nothing "heuristic" about this approach. A hyperbolic trajectory is a trajectory of the fluid flow having stable and unstable manifolds. The stable and unstable manifolds are made up of trajectories, this is why trajectories cannot cross them. They ARE flow barriers by construction (they do not have to be "identified"). In other words, they are the direct construction of Lagrangian coherent structures. FTLEs and Lagrangian descriptors are methods to detect these structures (not to construct them).

In several places the authors use the word "ridges" to refer to some property of Lagrangian descriptors. It is not clear what they mean by this since, to my knowledge, it has not been used in the literature to refer to any property of Lagrangian descriptors. It was used in the original Shadden and Marsden paper to refer to a feature of FTLE fields, and in that paper it was given a precise mathematical meaning. However, that meaning appears to have largely been "lost" as people now tend to throw around the term rather cavalierly (as

these authors have done) without providing an understanding of its context and mathematical definition in the situation in which they are writing.

In Section 4.2 the authors add noise to their velocity field. The details of this are not clear, especially if, and how, Lagrangian descriptors would fit into this framework. Lagrangian descriptors are integrations of positive quantities over trajectories. Noisy velocity fields give rise to stochastic ODEs, whose solutions are stochastic processes, not trajectories.

In summary, this paper should not be published. The authors do not exhibit a clear understanding of the topics on which they are writing. The paper contains fundamental misunderstandings, and it displays a profound lack of knowledge of the relevant literature. The section on the lifetimes of eddies was interesting. However, because of the way in which the paper was written in is not clear what of the rest of the paper is required for that work.

---

## Short Comment (SC2) · 10 Mar 2016

A scalar field is objective if it is independent of the observer (see introductory books on continuum mechanics). Specifically, an objective scalar field remains unchanged under all Euclidian coordinate changes of the form

$$x = Q(t)y + b(t), \quad (*)$$

where x are the coordinates in the present frame, y are the coordinates in the new frame, Q(t) is a proper orthogonal tensor and b(t) is a vector. Both Q and b should be continuously differentiable in time, but otherwise arbitrary. A weaker notion of frame independence is Galilean invariance, which only requires the above invariance for

Q(t)\equiv 0, i.e., only under translations of the frame but not under rotations.

An axiom of continuum mechanics is that material response (including material transport) is an objective phenomenon and hence can only be described self-consistently via objective quantities. Indeed, the assessment of whether a fluid mass travels coherently or mixes with its environment should not be dependent on coordinates (I've just described it without coordinates). A person on a boat, a circling airplane or a ship should invariably reach the same conclusion in this regard. For this reason, non-objective coherent structure detection methods cannot capture material transport self-consistently.

The M-function (the length of trajectories in a velocity field) is not objective, given that the trajectory length depends on the observer. For instance, for an observer traveling with any given trajectory of a vector field, the length of that trajectory is zero. So, by a simple Galilean change of the frame, the M function can be made zero at any desired location. This simple argument proves that the M-function is not even Galilean invariant, let alone objective.

One might still contend, however, that at least the topological features (say, maxima and minima) of the M function are objectively defined, even if its values change from one frame to the other. That is not true either, unfortunately.

Indeed, take a trivial flow that is just full of fixed points (a fluid at rest) in the x-frame. The M function is, therefore, identically zero, suggesting (correctly) that there is no vortex in this steady flow. Pick now an arbitrary constant vector b(t) \equiv b_0, and any rotation matrix Q(t). Then in the y-frame defined by the coordinate change (*), the location y=0 (formerly x=b_0) has zero trajectory length, whereas all other points in the y-frame will be rotating around b_0, accumulating nonzero trajectory length. The angular velocity of this rotation is the same for all these points (governed by Q(t)), so the further these points are from b_0, the larger arclength they cover in a given period of time. Consequently, in the y frame, the M function has a global minimum around the

point x=b_0 that I have just arbitrarily selected above.

Before anyone says "... but it did give the right answer in the original frame!", consider that you would not know which of the two different answers to trust, had I not given you the answer in advance. I could have started by giving you the velocity in the y frame, and you would have given me the wrong answer in the original frame. In a truly unsteady flow, there is no distinguished frame [Lugt, 1979].

Therefore, the topology of the M function (including the locations of minima) is not objective either. Anyone thinking of defining a vortex/eddy as a region filling a valley surrounding a local minimum of the M function should keep the above example in mind.

Without doubt, under Lagrangian advection, any scalar quantity will create patterns when plotted over initial conditions. That's precisely why coherent structures are important: they tend to create coherent pattens in everything advected by the flow. But this does not imply that the advected quantity (including the creamer in one's stirred coffee or a piece of fishnet in the ocean) has any deep, intrinsic meaning for coherent structures in the carrier fluid. More pseudo-mathematically speaking, one can certainly integrate $x\_1^3+15.4*x\_2$ over trajectories and plot the results over the initial conditions of those trajectories in the $(x\_1,x\_2)$ plane. One might then ponder about the deep meaning of this cubic polynomial for fluid transport, encouraged by the various patterns that will undoubtedly emerge (unless the flow is a parallel shear flow in the $x\_2$ direction).

Yet perhaps most would agree at this point: this cubic polynomial has no meaning for fluid mixing. To make this point, one does not need to engage in an endless argument with the proud inventor of this cubic diagnostic (that would be myself), who will no doubt defend this great tool, saying that critiques simply do not understand the method. Instead, one can simply point out that this diagnostic is not objective and hence cannot possibly capture anything intrinsic about material transport. End of discussion.

The continuum mechanics community went through the same deliberation a long time

ago. At some point, they stopped even considering newly proposed heuristic constitutive laws if those laws were not objective. It might be time for this to happen in the coherent structure detection industry as well.

---

## Referee Comment (RC2) · K. McIlhany (Referee) · 15 Mar 2016

Several typographical errors to correct: 1) Overall, many missing commas. When starting a sentence with a prepositional phrase, separate it from the sentence with a comma. Page/Line 1/13 - For this reason, 7/11 - For the Lagrangian descriptor $M_v$, 7/16 - ... in a flow, 9/26 - For $M_v$, 14/4 - However, nowadays, 14/12 - However,

In some cases, there are commas that are unnecessary, page 5 line 8: "...dynamical evolution yield" (no comma).

Do a re-read of the paper and look for these prepositional phrases and clearly separate them grammatically.

[Figure]

2) At times, the papers language is too conversational. In general, the tone of the paper is scientific, and it should remain so throughout the paper. From above, "nowadays" is an example. Page/Line 3/15 - "Anyhow" can simply be removed. 7/6 - remove "Again".

Do a re-read and it helps to read it out loud so that you can catch the conversational tone when it comes up. That said, the language is outstanding for a non-English native speaker!

3) Several words can be removed as they are unnecessary. In some cases, words need to be added or changed. Page/Line 1/14 - change "e.g. marine biology." to "marine biology for instance." 2/21 - "...but have been recently..." 4/31 - change on to of 5/10 - Change: "Manifold trajectories on both sides of the manifold have different behaviors compared..." 8/6 - change "distinction" to "distinguishing between ... and the identification " 9/2 - change "We use its" to "We use the feature" 14/3 - Change "non" to "none"

Technical issue

4) Check your formulae: Page/Line 5/3 - equation #3 - make sure the velocity is squared and the dt is not under the sqrt

Overall technical comments:

My main problem with this paper is that it asserts things that it does not support directly in the text. At times, there are conflicting statements about what the newly proposed Euler-Lagrangian descriptor can and cannot do. These discrepancies need to resolved in the text so that the reader is not confused or led astray. Also, in the beginning of the paper, the use of oceanographic data is discussed, but the paper is essentially about toy-models. I understand the need to verify a new metric by using toy-models, however, if you suggest that this metric is useful for geo-physical flows, then you need to demonstrate that in this paper, or put a disclaimer early within the text, that you intend to follow-up this paper with another paper demonstrating the metric on actual geophysical flows obtained from either satellite data or well-understood simulated oceano-graphic models such as CCSM4 or a variant of ROMS that is well-accepted as a good representation of historical data (flows). Finally, when you compare your new metric to existing metrics, then state they your metric is better, you need to clearly state the differences and exactly HOW your metric out performs another. That is simply now done well in the text as it currently stands.

From my understanding of $M_v$, you state its superiority over the M-value mainly because it maximizes when a fluid packet is part of gyre. In this case, for the duration of its stay within the gyre, the vorticity is high so the $M_v$ will be maximal. For the M-value, the center of the gyre will be a minimum, such that the $M_v$ can distinguish an elliptical point as well as a hyperbolic point, whereas, the M-value shows both types as minima. That is the main difference you quote in their behavior.

First, you state that your metric has excellent time resolution when seeking the beginning of a gyres formation as well as its lifetime, because you can measure when the gyre dies off. In both of these measurements, you depend on the value of tau. You make a cases in figure 6 that the best value for tau is 0.15 times the lifetime of the gyre. This is a circular definition. You need to know the lifetime in order to determine tau if it is to be based on a percentage of that lifetime. Furthermore, you can only find a gyre once you vorticity values are maximized, meaning that you need a particle to have already been inside of a gyre long enough for the $M_v$ to become maximal. This means that there is a lead-in time where you do not know whether you are in a gyre or not as the trajectory has not had enough time to sample to gyre. The problem is that in order to find the gyre in the first place, you need an initial value for tau simply to compute the $M_v$. So, do you propose to constantly be computing $M_v$ for a range of tau values until you find a gyre - THEN you can adjust tau to be 0.15 the lifetime of the gyre? But wait, you need to know the end of the gyre as well to know the lifetime, so you cannot determine an optimal tau to find a gyre until it has formed and gone away. This suggests that an oceanographer will need to be computing $M_v$ over a range of
tau values constantly simply to see when/if a gyre has formed. Of course, this is also true for the M-value.

Figure 3 needs to be larger and with a better color contrast in order to show the manifold structure.

Page 5, lines 6-15. At the beginning of the paragraph, you state that the M-value can distinguish between stable and unstable manifolds as well as hyperbolic and elliptic regions. On line 14, you state that the M-value cannot distinguish between elliptic and hyperbolic points.

Page 7, figure 2. It is implied in the previous literature as well as your own figures, that the M-value is good at finding the radius of the elliptic regions BECAUSE it has a minimum as the center, so that the contour of M-value maximizes as it moves away from the center and then decays as it moves far away - such that the maxima of M-values could be used to estimate the radii of elliptic regions. This is not explained in your paper, yet, you regularly refer to needing to use both the M_v and M-value to extract useful gyre information. Page 9, lines 3 and 4 - refer to using the M_v in combination with the M-value. Pages 15-17 also make it unclear in all of the figures which M function is used to extract the gyre location AND SIZE. In the figures, is it stated M and M_v. Why both?

Page 8, lines 6-10. This paragraph asserts that M_v is the best of four metric because it can discern between stable and unstable manifold lines - which can be used to get more insight into the size of the eddies. HOW exactly? I feel like a paragraph explaining this statement is missing. Perhaps it would precede this paragraph. Can M_v distinguigh between stable and unstable manifold lines? If so, how? For that matter, in Figure 2, you show the four convective cell case, where M_v maximizes at the center. As tau increases, the maxima form a flatter and flatter plane centered on the gyre. Doesn't this make you less sensitive to the size of the gyre, not more sensitive? How does M_v determine the radius of a gyre. I'd like to know based on the text provided.

Page 11, figure 6. The resolution shown for this figure does not convince me that 0.15*lifetime is the optimal tau value. It could be any value from 0.06 up to 0.21*lifetime. There should be many more points to determine the best value.

Finally, in the beginning, I thought I was going to see this metric applied to an oceanographic data set. By the end, I did not find it. Please show me something geo-physical or tell me that it is coming in a later publication.

Conclusion:

I do find the approach taken opens up a path to many Eulerian-Lagrangian metric to be devised. This paper could serve as a warning to others about the nuances required to create and utilize such a metric. There is something new here, however, the case currently is weakened by gaps in the presentation that lead to more questions in the readers mind. I look forward to the authors filling in these gaps and then publishing!

---

## Referee Comment (RC3) · Anonymous Referee #3 · 20 Mar 2016

The authors consider the classical problem of detecting and tracking eddies in flow fields (in the title the adjective 'oceanic' is present, but the paper is about kinematic flows). To do so they develop a variant, vorticity-based, of the so-called 'Lagrangian descriptors', and evaluate it in model kinematic flows, as compared with other Lagrangian and Eulerian methodologies. There is some interesting material there, but in my opinion, the paper in its present form does not achieve the quality level required to recommend publication in NPG. In the following I summarize the main points that, in my opinion, would require significant revision:

- There is a huge literature on the problem of eddy detection, coming from very different scientific communities. Thus it is increasingly complicated to do something really

new and to do justice to the vast literature. I should recognize, however, that the authors do a reasonable summarizing job in their introduction. Unavoidably, there are important recent results missing. From the part of the literature I know, I feel the following two references merit some citation and discussion: Karrasch D, Huhn F, Haller G. 2015 Automated detection of coherent Lagrangian vortices in two-dimensional unsteady flows. Proc.R.Soc.A 471: 20140639. http://dx.doi.org/10.1098/rspa.2014.0639 Haller G., Hadjighasem A, Farazmand M, Huhn F Defining Coherent Vortices Objectively from the Vorticity http://arxiv.org/abs/1506.04061

- There is a number of imprecise or even false statements in the paper. Here is a selection of them: * p. 2, lines 26-28: It is stated that algorithms to find DHT rely on 'Lagrangian descriptors'. Please note that DHTs were defined and computed many years before the introduction of the Lagrangian descriptors. * p. 2, line 31: This sentence makes no sense: 'The unstable manifolds are often called material lines in 2d () and surfaces in 3d flows ()' * In many places the authors use the word 'fixed point' for what are special elliptic or hyperbolic trajectories (moving, and then not fixed at all): abstract, pages 5, 6, 7, 8, 9, 15, 17 ... this is deeply misleading.

- In Mancho et al 2013 it is clearly stated that essentially any fluid property can be integrated along trajectories and provide a suitable 'Lagrangian descriptor'. In this sense the use of the vorticity is just another example of 'Lagrangian descriptor'. I find the name 'Euler-Lagrangian descriptor' and the emphasis given in the discussions to the mixed character rather inadequate.

- I hardly can see any 'manifold' in the plots of M and specially of M_v in Fig. 3. Perhaps tau=0.15 is too small, or the contrast of the figure is not enough.

- At a first sight it looks incorrect to say that M, at variance with M_v, can not distinguish between elliptic and hyperbolic areas, since in any plot of M one can clearly identify them. But after some thinking I recognize that there is a real advantage (perhaps the only one) of M vs M_v, which is the fact that ellipticity and hyperbolicity are

simply assessed by the maximum or minimum character of M_v, much more easy to automatize that the more complex neighbourhood exploration needed for the case of M. But then I do not understand (and the authors do not give any hint of it) why in Section 4 they say they need a combination of M_v and M, instead of just M_v.

- I think that the most original part of this research is the assessment of the behaviour of the different indicators under different types of noise. Nevertheless, the definitions of noise types in page 11 are all incomplete: for type 1 and 2 one can not reproduce the paper results unless the authors define 'noise strength', given that for white noise this would depend on the particular spatial and temporal discretizations used, which are not completely stated. For type 3, it is only after reading a comment in the Supplemental material that one begins to understand that noise is added to the functions h1 and h2, but again, 'strength' or 'noise level' should be properly defined.

- In the Supplemental material, Sect. S1 there is no indication on how the time-dependence, needed to define T_c, is introduced in the seeded eddy model. Also I find very convoluted (and not well explained) the way the radii of the eddies are sampled. Since at the end the authors restrict to 15-25 km radii, it seems to me that all this complexity is irrelevant and that anything uniform or Gaussian in that range will give the same results.

- Errata: * There is a missing square of the velocity in Eq. (3) * Page 12, line 14: signal to noise ratio small? or large?

In summary, I do not recommend publication of the paper, and recommend extensive revision.

---

## Editor Comment (EC1) · A. M. Mancho (Editor) · 7 Apr 2016

Some remarks are required on the comment "Non-objectivity of the M function and other thoughts" by G. Haller, as it contains several inaccurate and misleading statements. The comment argues that the M-function is non-objective because it provides different outputs for dynamical systems that are related by a Galilean coordinate transformation. In particular the comment discusses in detail a particular example for which claims that the output of M in one of the frames is wrong. The comment considers two 2D autonomous dynamical systems that are related by a Galilean transformation. The quoted example is described in fact in the basic dynamical system book *Ordinary Differential Equations* by Arnold (p. 44, Problem 3 and p. 45, Fig. 47) and for the

system in which M is claimed to provide the 'wrong' result, M provides in reality exactly the same description which is found in this book and never questioned until the above comment. Despite being related by a Galilean transformation the phase space structure of the two dynamical systems is very different. Therefore the M function should provide different information in each case–information that reflects the phase space structure for the particular dynamical system. More details for this example are given next.

In particular the comment considers the following dynamical system:

$$\dot{x} = 0, \qquad \text{where} \quad x \in \mathsf{IR}^2 \tag{1}$$

We subject this vector field to a Galilean transformation, i.e., a rotation $y = R(t)^T x$, such that in the rotating frame sys1 has the form:

$$\begin{aligned} \dot{y}_1 &= y_2 \\ \dot{y}_2 &= -y_1. \end{aligned} \tag{2}$$

Here $R(t)^T$ is the transpose of the orthogonal matrix:

$$R(t) = \begin{pmatrix} \cos t & -\sin t \\ \sin t & \cos t \end{pmatrix}$$

The phase portrait of (1) consists entirely of fixed points. The phase portrait of (2) consists of a one-parameter family of invariant circles. It will be convenient to express (2) in action angle variables $(\rho, \theta)$ as follows:

$$\dot{\rho} = 0 \qquad (3)$$
$$\dot{\theta} = -1$$

where $y_1 = \rho \cos \theta$ and $y_2 = \rho \sin \theta$. $H$ is the Hamiltonian in action-angle variables: $H(\rho, \theta) = \rho$. From these expressions it is clear that $(\rho = \rho_0, \theta(t) = -t + \theta_0)$ are solutions to the system (3) which correspond to invariant 1-tori.

Now we apply the M function to (1) and (3), where M measures the arclength of a trajectory through an initial condition in both forward and backward time. Clearly, for (1) M is zero for all initial conditions since every point is a fixed point (see Madrid and Mancho (2009)). For (3) M=$(2\tau)\rho$ where $2\tau$ is the forwards-backwards time interval length. Hence the contours of $M$ are in 1-1 correspondence with the trajectories of (3) (see Mezic and Wiggins (1999) and Susuki and Mezic (2009)). Hence M recovers the correct phase space structure for both (1) and (3). If M were the same for both of these vector fields it would not accurately recover the phase space structure for each vector field, as is the case of the Lagrangian-averaged vorticity deviation (LAVD).

Please also note the supplement to this comment:
http://www.nonlin-processes-geophys-discuss.net/npg-2016-16/npg-2016-16-EC1-supplement.pdf
* * *

---

## Author Comment (AC1) · 18 May 2016

Response to Reviewer #2

We would like to thank the reviewer for his encouragement and positive assessment of our manuscript and the suggestions to improve the manuscript, which we have taken into account. Here we respond to all the comments made by the reviewer and indicate the changes in the manuscript made accordingly.

*Several typographical errors to correct: 1) Overall, many missing commas. When starting a sentence with a prepositional phrase, separate it from the sentence with a comma. Page/Line 1/13 - For this reason, 7/11 - For the Lagrangian descriptor M_v, 7/16 - ... in a flow, 9/26 - For M_v, 14/4 - However, nowadays, 14/12 - However,*

*In some cases, there are commas that are unnecessary, page 5 line 8: "...dynamical evolution yield" (no comma).*

*Do a re-read of the paper and look for these prepositional phrases and clearly separate them grammatically.*

We have done our best in re-reading the manuscript and improving grammar. Since we are not native English speakers, we admit having some problems with that.

*At times, the papers language is too conversational. In general, the tone of the paper is scientific, and it should remain so throughout the paper. From above, "nowadays" is an example. Page/Line 3/15 - "Anyhow" can simply be removed. 7/6 - remove "Again".*

*Do a re-read and it helps to read it out loud so that you can catch the conversational tone when it comes up. That said, the language is outstanding for a non-English native speaker!*

Thank you very much for pointing out the conversational language. We have removed it, whenever we noticed it ourselves.

*Several words can be removed as they are unnecessary. In some cases, words need to be added or changed. Page/Line 1/14 - change "e.g. marine biology." to "marine biology for instance." 2/21 - "...but have been recently..." 4/31 - change on to of 5/10 - Change: "Manifold trajectories on both sides of the manifold have different behaviors compared..." 8/6 - change "distinction" to "distinguishing between ... and the identification " 9/2 - change "We use its" to "We use the feature" 14/3 - Change "non" to "none"*

We have changed the text accordingly.

*Check your formulae: Page/Line 5/3 - equation #3 - make sure the velocity is squared and the dt is not under the sqrt*

We have corrected the formula.

*My main problem with this paper is that it asserts things that it does not support directly in the*

*text. At times, there are conflicting statements about what the newly proposed Euler-Lagrangian descriptor can and cannot do. These discrepancies need to resolved in the text so that the reader is not confused or led astray. Also, in the beginning of the paper, the use of oceanographic data is discussed, but the paper is essentially about toy-models. I understand the need to verify a new metric by using toy-models, however, if you suggest that this metric is useful for geo-physical flows, then you need to demonstrate that in this paper, or put a disclaimer early within the text, that you intend to follow-up this paper with another paper demonstrating the metric on actual physical flows obtained from either satellite data or well-understood simulated oceanographic models such as CCSM4 or a variant of ROMS that is well-accepted as a good representation of historical data (flows).*

We agree completely with the reviewer that a demonstration of the method with a real oceanographic velocity field is much more convincing. To apply the Lagrangian descriptor $M_v$ based on the modulus of vorticity and the eddy tracking employing it to an oceanographic velocity field, was already planned when we submitted the manuscript. Now, we have included an example for the western Baltic Sea in the revised version and replaced the Section about the seeded eddy model. Furthermore, we discuss on the basis of this example advantages and disadvantages of the method if it is applied to an oceanographic data set. The velocity field for the western Baltic Sea is from the ocean model described in Gräwe et al. (2015a). Further research aims at an eddy statistics for lifetime, size and track of eddies for the central Baltic Sea with the eddy tracking based on $M_v$. But this work is beyond the scope of this current manuscript.

*Finally, when you compare your new metric to existing metrics, then state they your metric is better, you need to clearly state the differences and exactly HOW your metric out performs another. That is simply now done well in the text as it currently stands.*

We have rewritten and complemented the parts of the text where we explain what the new metric searches for and what are the differences to existing methods. We hope it is now easier to understand and more precise. We now discuss in more detail the problems arising when applying this metric to a real oceanographic field. This sheds more light on the difficulties of an automated eddy detection. The comparison of the results obtained with $M_v$ and the eddy tracking toolbox by Nencioli et al. (2010) reveals that for both methods false positives and false negatives exist. To improve those results is a future challenge.

*From my understanding of $M_v$, you state its superiority over the M-value mainly because it maximizes when a fluid packet is part of gyre. In this case, for the duration of its stay within the gyre, the vorticity is high so the $M_v$ will be maximal. For the M-value, the center of the gyre will be a minimum, such that the $M_v$ can distinguish an elliptical point as well as a hyperbolic point, whereas, the M-value shows both types as minima. That is the main difference you quote in their behavior.*

*First, you state that your metric has excellent time resolution when seeking the beginning of a*

*gyres formation as well as its lifetime, because you can measure when the gyre dies off. In both of these measurements, you depend on the value of tau. You make a cases in figure 6 that the best value for tau is 0.15 times the lifetime of the gyre. This is a circular definition. You need to know the lifetime in order to determine tau if it is to be based on a percentage of that lifetime. Furthermore, you can only find a gyre once you vorticity values are maximized, meaning that you need a particle to have already been inside of a gyre long enough for the M_v to become maximal. This means that there is a lead-in time where you do not know whether you are in a gyre or not as the trajectory has not had enough time to sample to gyre. The problem is that in order to find the gyre in the first place, you need an initial value for tau simply to compute the M_v. So, do you propose to constantly be computing M_v for a range of tau values until you find a gyre - THEN you can adjust tau to be 0.15 the lifetime of the gyre? But wait, you need to know the end of the gyre as well to know the lifetime, so you cannot determine an optimal tau to find a gyre until it has formed and gone away. This suggests that an oceanographer will need to be computing M_v over a range of tau values constantly simply to see when/if a gyre has formed. Of course, this is also true for the M-value.*

The proper choice of tau is indeed the main problem with any Lagrangian descriptor including M_v and M. Hence, for a real oceanographic problem one has to vary tau to find all the eddies. This necessary choice is a practical limitation of the method. We point to that fact now better in the manuscript and provide an improved figure to show the dependence on tau (Fig. 6).

*Figure 3 needs to be larger and with a better color contrast in order to show the manifold structure.*

We have changed the colorcode to improve the contrast, because a larger tau does not lead to a clearer structure. Unfortunately, the colorcode does not take into account colour-blindness, but we did not find any colorcode with enough color-dimensions that is also valid for color-blindness.

*Page 5, lines 6-15. At the beginning of the paragraph, you state that the M-value can distinguish between stable and unstable manifolds as well as hyperbolic and elliptic regions. On line 14, you state that the M-value cannot distinguish between elliptic and hyperbolic points.*

Manifolds as well as hyperbolic and elliptic fixed points (more general distinguished hyperbolic trajectories and distinguished trajectories surrounded by an elliptic region in the sense of Mancho et al. (2013) correspond to singular features in the plot of M (singular lines and local minima). In this sense M can identify them. We apologize for the misleading use of the term "distinguish" on page 5 line 6 it was meant in the sense of "identify" (We are not native speakers.). We have rewritten the section on the Lagrangian descriptor M to make clear that M can of course identify distinguished hyperbolic trajectories and distinguished trajectory surrounded by an elliptic region but they are both displayed as a local minimum of M from which one cannot decide if it has elliptic or hyperbolic properties. Therefore, we constructed a vorticity based Lagrangian descriptor M_v that yields singular lines and local minima and

maxima where the local maxima correspond to eddy cores (moving elliptic points) and the local minima to the distinguished hyperbolic trajectories ("moving saddle point").

*Page 7, figure 2. It is implied in the previous literature as well as your own figures, that the M-value is good at finding the radius of the elliptic regions BECAUSE it has a minimum as the center, so that the contour of M-value maximizes as it moves away from the center and then decays as it moves far away - such that the maxima of M- values could be used to estimate the radii of elliptic regions. This is not explained in your paper, yet, you regularly refer to needing to use both the M_v and M-value to extract useful gyre information. Page 9, lines 3 and 4 - refer to using the M_v in combination with the M-value. Pages 15-17 also make it unclear in all of the figures which M function is used to extract the gyre location AND SIZE. In the figures, is it stated M and M_v. Why both?*

As explained above M and M_v yield singular features (singular lines and local minima and maxima).

The eddy core in case of M corresponds to a local minimum and in case of M_v to a local maximum. Because the Lagrangian descriptor M would display a minimum in case of a DHT too a second criterion is needed to distinguish them properly. Therefore, we suggest M_v to simplify the automated eddy detection because one has only to search for a local maximum that corresponds to the eddy core.

The local maxima and the singular lines of M_v will be used to construct an eddy tracking tool based on the following concept of an eddy: We denote an eddy as being bounded by pieces of stable and unstable manifolds of DHTs (according to Branicki et al. (2011) and Mendoza and Mancho (2012)) surrounding an area in which the flow is rotating. The manifolds correspond to singular lines in M_v which are used to describe the eddy boundaries. The eddy core is considered as a local maximum of M_v within this bounded region, which can be interpreted as one point of a distinguished trajectory surrounded by an elliptic region.

For the detection of the eddy shape we have previously used a combination of M and M_v because M shows in our test case a clear line of minimum M values that was easier to detect automated than the line in M_v. In general, manifolds correspond to singular lines (Mancho et. al. 2013). To construct an eddy shape detection that is more general and only based on M_v, we have improved the shape detection algorithm. The improved shape detection is based on the assumption that the eddy boundary is the largest closed contour line of M_v where M_v is an extremum (large gradient of M_v).

Furthermore, we have rewritten the Sections 2 and 3 to clarify the idea of M and M_v and its correspondence to our understanding of an eddy.

*Page 8, lines 6-10. This paragraph asserts that M_v is the best of four metric because it can discern between stable and unstable manifold lines - which can be used to get more insight into the size of the eddies. HOW exactly? I feel like a paragraph ex- plaining this statement is*

*missing. Perhaps it would precede this paragraph.*

*We have rewritten this paragraph and parts of Sect. 2 to make clear what the idea of the description of an eddy boundary based on manifolds is, namely to describe a region that is separated from the rest of the flow (as explained above and in* Branicki et al. (2011) and Mendoza and Mancho (2012)*).*

*Can M_v distinguish between stable and unstable manifold lines? If so, how?*

Singular lines in the plot of M or M_v correspond to manifolds. But one cannot distinguish based on the plot of M or M_v if it is a stable or unstable manifold. For the understanding of an eddy as a region bounded by pieces of stable and unstable manifolds of the distinguished hyperbolic trajectory ("moving saddle point") with an eddy core inside, it is only necessary to identify the manifolds and not the type of the manifold. Furthermore, if one is interested in the type of the manifold one can put tracers on the manifold close to the hyperbolic trajectory and track them forward and backward in time.

*For that matter, in Figure 2, you show the four convective cell case, where M_v maximizes at the center. As tau increases, the maxima form a flatter and flatter plane centered on the gyre. Doesn't this make you less sensitive to the size of the gyre, not more sensitive? How does M_v determine the radius of a gyre. I'd like to know based on the text provided.*

The maximum of M_v does not form a flatter and flatter plane in the centre, instead the centre becomes sharper and sharper as minimum of M in figure 2 f). This cannot be seen so clear in the colorcode used because the maximum is a light yellow point in a yellowish region. As mentioned above we have changed the colorcode to improve the contrast.

*Page 11, figure 6. The resolution shown for this figure does not convince me that 0.15\*lifetime is the optimal tau value. It could be any value from 0.06 up to 0.21\*lifetime. There should be many more points to determine the best value.*

We have improved the figure and calculated more values. The chosen value of tau=0.15\*lifetime is in our case the beginning of a 0.02 small region of the optimal tau values. We have chosen the lower bound of this region to minimize the computational effort for calculating M_v.

*Finally, in the beginning, I thought I was going to see this metric applied to an oceano- graphic data set. By the end, I did not find it. Please show me something geo-physical or tell me that it is coming in a later publication.*

We have applied the method to an example of the western Baltic Sea to give an outlook on the application to oceanographic data sets.

---

## Author Comment (AC2) · 18 May 2016

Response to Reviewer #1

We would like to thank the reviewer for his critical assessment of our manuscript and the suggestions to improve it, which we have taken into account. In the following we respond to all the concerns of the reviewer and indicate the changes in the manuscript:

*The authors develop a new Lagrangian descriptor based on integrating the magnitude of the vorticity (an Eulerian quantity) along trajectories. They claim this is a new Eulerian-Lagrangian descriptor". However, their approach follows exactly the methodology for Lagrangian descriptors described in the following*
*paper:*
*A. M. Mancho, S. Wiggins, J. Curbelo, C. Mendoza Lagrangian descriptors: A method for revealing phase space structures of general time dependent dynamical systems, Communications in Nonlinear Science and Numerical Simulation, 18(12), 3530 3557 (2013)*
*At the bottom of page 3532 of this paper it is stated that Lagrangian descriptors can be constructed by integrating any bounded positive intrinsic physical or geometric property of the velocity field...". Certainly the magnitude of the vorticity satisfies that criteria, but also the magnitude of the velocity field, whichis a common Lagrangian descriptor used in the abovementioned paper (henceforth, Mancho et al 2013). It just so happens that the integral of the square root of the magnitude of the velocity field along trajectories has the interpretation of arclength, but it is still of the same character as the quantity studied bythe authors who wish to rename the quantity Eulerian-Lagrangian descriptor". This is completely misleading and contrary to the methodology introduced in Mancho et al 2013. Indeed, all of the quantities in Mancho et al. 2013 would then be `Eulerian-Lagrangian descriptors" in the terminology of the authors of the paper under review as all of the quantities of Table 1 in Mancho et al. 2013*
*proposed for the construction of Lagrangian descriptors are Eulerian quantities. In this sense all Lagrangian descriptors are constructed from Eulerian and Lagrangian quantities, with the purpose of providing Lagrangian transport information. So there is nothing new methodologically" in this paper.*

We completely agree with the reviewer and also reviewer #3 that already Mancho et al. 2013 pointed out, that any fluid property can be used to construct a Lagrangian descriptor. This has been pointed out by us explicitly already in the first version of the manuscript (cf. the sentence "As already pointed out by Mancho et al. (2013) any intrinsic physical ..." in the beginning of the paragraph before formula (4)). Our motivation to introduce the name Euler-Lagrangian descriptor was a more practical one. Because we found it difficult to read talking about one and another Lagrangian descriptor, we introduced a distinction by the names Euler-Lagrangian for one of them and Lagrangian for the other. Since this has been found misleading by two reviewers (#1 and #3) because it would look like the definition of a new descriptor, which indeed is not the case as we of course know, we have changed this in the revised version. To emphasize this we have now avoided the name Euler-Lagrangian descriptor in the title and throughout the whole manuscript. Furthermore, we have even more than before emphasized that the original idea had been already formulated in Mancho et al. 2013. We now write "We would like to emphasize, that it has been already pointed out by Mancho et al. (2013)".

*The authors claim that Lagrangian descriptors are not objective, and justify this claim by referring to reference [15]. However, we have looked at reference [15], and I cannot find any proof of non-objectivity for Lagrangian descriptors in that reference. If the authors are going to make such a strong claim, then they must provide a reference to a proof of their claim.*

We have removed the word heuristic as well as the whole discussion about objectivity from the text to avoid any further discussion of this issue since it is not the aim of this manuscript to clarify a question, which is debated in the literature. Our aim is much more practical and does not claim to develop a new mathematical method. Since the reviewer does not find Ref. [15] appropriate, we are thankful to George Haller to provide his comment #2 containing the arguments requested. Ana Mancho has answered the question of objectivity of Lagrangian descriptors in an editor's comment. The main focus of this manuscript is completely different and does not concentrate on the question whether the method is objective or not. It is just an application of an existing method to tackle the question of providing a robust method to detect and count eddies in an oceanographic flow. We regret that the problem of objectivity distracted the reviewer from the main focus of the manuscript. However, since this problem has been addressed in several comments, we cannot ignore it and refer the reader now to this discussion in the discussion section of the journal.

*The authors are proposing what they refer to as a new characterization of eddies based on an elliptic region bounded by segments of stable and unstable manifolds of a hyperbolic trajectory. This allows the lobe dynamics mechanism to control transport in and out of the elliptic region. However, a careful development of eddies from this point of view has already been given in the following references:*
*M. Branicki and S. Wiggins, Finite-time Lagrangian transport analysis: stable and unstable manifolds of hyperbolic trajectories and finite-time Lyapunov exponents, Nonlin. Processes Geophys., 17, 136 (2010).*
*M. Branicki, A. M. Mancho, and S. Wiggins, A Lagrangian description of transport associated with a front-eddy interaction: Application to data from the North-Western Mediterranean Sea, Physica D, 240(3), 282-304, (2011).*

We would like to thank the reviewer for pointing out those papers to us, which we now cite in the text.

*The authors claim that Lagrangian descriptors cannot detect eddies. I find this to be a very surprising statement based on several papers in the literature that characterize eddies in the Gulf stream and Gulf of Mexico, for example, in terms of Lagrangian descriptors.*

We did not claim in the manuscript that Lagrangian descriptors cannot detect eddies. We have only written that the Lagrangian descriptor using the path length identifies both the eddy core and the DHTs with a minimum of M. This is a bit cumbersome when designing an algorithm for which it is impossible to check for each whether it corresponds to a hyperbolic point or elliptic point (eddy core). Therefore, we were looking for another Lagrangian descriptor not posing this difficulty. Using the Lagrangian descriptor $M_V$ has the advantage that no distinction is needed between eddy cores and DHTs, since they are displayed as maxima and minima of $M_V$ respectively. By contrast, the Lagrangian descriptor M needs an additional criterion since eddy cores and DHTs are both displayed as minima. Since the reviewer misunderstood our statements we have now reformulated it in a more precise way.

*The word "heuristic" and the phrase "to identify Lagrangian coherent structures in a flow" are used very bizarrely, and incorrectly, here. First, there is nothing "heuristic" about this approach. A hyperbolic trajectory is a trajectory of the fluid flow having stable and unstable manifolds. The stable and unstable manifolds are made up of trajectories, this is why trajectories cannot cross them. They ARE ow barriers by construction (they do not have to be "identfied"). In other words, they are the direct construction of Lagrangian coherent*

*structures. FTLEs and Lagrangian descriptors are methods to detect these structures (not to construct them).*

According to our previous answer to the same criticism above we have deleted the word heuristic.
The word "identified" was used in the meaning that the manifolds are visible in the plot of M. We apologize the misleading use of the word. We are not native speakers.

*In several places the authors use the word "ridges" to refer to some property of Lagrangian descriptors. It is not clear what they mean by this since, to my knowledge, it has not been used in the literature to refer to any property of Lagrangian descriptors. It was used in the original Shadden and Marsden paper to refer to a feature of FTLE fields, and in that paper it was given a precise mathematical meaning. However, that meaning appears to have largely been "lost" as people now tend to throw around the term rather cavalierly (as these authors have done) without providing an understanding of its context and mathematical definition in the situation in which they are writing.*

Indeed, as the reviewer pointed out, the word „ridges" has been used in the context of FTLE fields, not only in the paper by Shadden and Marsden, but also in the papers by the group of Hernandez-Garcia/Lopez and coworkers from Spain. It just describes the local maxima of a certain quantity under investigation. We have deleted the word ridges in the context of Lagrangian descriptors.

*In Section 4.2 the authors add noise to their velocity field. The details of this are not clear, especially if, and how, Lagrangian descriptors would fit into this framework. Lagrangian descriptors are integrations of positive quantities over trajectories. Noisy velocity fields give rise to stochastic ODEs, whose solutions are stochastic processes, not trajectories.*

The reviewer is correct in saying that one hast o study stochastic differential equations when dealing with noisy velocity fields. This would apply if one would be interested in a stochastic view of the problem. However, in the manuscript we look at this problem from a very different point of view. Since we are interested to design an algorithm searching and detecting eddies in a real velocity field from oceanography, we wanted to test the algorithm against noisy velocity field in the way, that the noise is observational noise or in other words measurement errors. For many problems, the velocity field will not be given as solutions of a numerical integration but from observational data, which are corrupted by measurement errors. Therefore, we just added noise to the computed velocity field. This approach enables us to "mimic" in a simple way a noise, which comes from errors in field measurements. This is at the same time a test, how a Lagrangian descriptor would respond to velocity fields which do not fulfill the mathematical criteria of a two-dimensional divergence free velocity field. We have now explained this approach in more detail in the text and have rewritten most of the noise section accordingly.

---

## Author Comment (AC3) · 18 May 2016

Response to Reviewer #3

We would like to thank the reviewer for his/her thorough analysis of our paper and the suggestions given to improve the manuscript. We have addressed all the concerns of the reviewer and have rewritten the manuscript accordingly. In the following we give detailed answers to the questions raised and indicate the changes in the manuscript made as a response to the reviewers suggestions.

*There is a huge literature on the problem of eddy detection, coming from very differ- ent scientific communities. Thus it is increasingly complicated to do something really new and to do justice to the vast literature. I should recognize, however, that the authors do a reasonable summarizing job in their introduction. Unavoidably, there are important recent results missing. From the part of the literature I know, I feel the following two references merit some citation and discussion: Karrasch D, Huhn F, Haller G. 2015 Automated detection of coherent Lagrangian vortices in two-dimensional un-steady flows. Proc.R.Soc.A 471: 20140639. http://dx.doi.org/10.1098/rspa.2014.0639 Haller G., Hadjighasem A, Farazmand M, Huhn F Defining Coherent Vortices Objectively from the Vorticity http://arxiv.org/abs/1506.04061*

We would like to thank the reviewer for pointing out these two recent publications. The first of them we already cited in the first version of our manuscript. We were not aware of the second paper, since it had not appeared yet in a peer-reviewed journal. In a comment to our manuscript, this paper was also pointed out to us and we have now cited it. Additionally, reviewer #1 suggested several other citations, which we have included too.

*There is a number of imprecise or even false statements in the paper. Here is a selection of them: * p. 2, lines 26-28: It is stated that algorithms to find DHT rely on 'Lagrangian descriptors'. Please note that DHTs were defined and computed many years before the introduction of the Lagrangian descriptors. * p. 2, line 31: This sentence makes no sense: 'The unstable manifolds are often called material lines in 2d () and surfaces in 3d flows ()' * In many places the authors use the word 'fixed point' for what are special elliptic or hyperbolic trajectories (moving, and then not fixed at all): abstract, pages 5, 6, 7, 8, 9, 15, 17 ... this is deeply misleading.*

We agree with the reviewer, that the concept of DHT has been developed earlier and have changed the corresponding formulation in the text.

That the unstable manifolds are often called material lines is taken from the literature, where one can find these formulations rather often. But according to the suggestion of the reviewer we have removed this sentence since it does not add to the content of our manuscript.

We thank the reviewer for pointing out the misleading formulation of fixed points in the manuscript. We have rewritten the whole text for the introduction of the Lagrangian descriptor to avoid any confusion. Indeed, when we were writing about fixed points we meant

indeed the elliptic and hyperbolic trajectories. We have changed that in the current version of the manuscript.

*In Mancho et al 2013 it is clearly stated that essentially any fluid property can be integrated along trajectories and provide a suitable 'Lagrangian descriptor'. In this sense the use of the vorticity is just another example of 'Lagrangian descriptor'. I find the name 'Euler-Lagrangian descriptor' and the emphasis given in the discussions to the mixed character rather inadequate.*

We completely agree with the reviewer and also reviewer #1 that already Mancho et al. 2013 pointed out, that any fluid property can be used to construct a Lagrangian descriptor. This has been mentioned by us explicitly already in the first version of the manuscript (cf. the sentence "As already pointed out by Mancho et al. (2013) any intrinsic physical ..." in the beginning of the paragraph before formula (4)). Our motivation to introduce the name Euler-Lagranian descriptor was a more practical one. Because we found it difficult to read talking about one and another Lagrangian descriptor, we introduced a distinction by the names Euler-Lagrangian for one of them and Lagrangian for the other. Since this has been found misleading by two reviewers (#1 and #3) because it would look like the definition of a new descriptor, which indeed is not the case as we of course know, we have changed this in the revised version. To emphasize this we have now avoided the name Euler-Lagrangian descriptor in the title and throughout the whole manuscript. Furthermore we have even more than before emphasized that the original idea had been already formulated in Mancho et al. 2013. We now write "We would like to emphasize, that it has been already pointed out by Mancho et al. (2013)…".

*I hardly can see any 'manifold' in the plots of M and specially of M_v in Fig. 3. Perhaps tau=0.15 is too small, or the contrast of the figure is not enough.*

We have changed the colorcode to improve the contrast, because a larger tau does not lead to a clearer structure. Unfortunately, the colorcode does not take into account colour-blindness, but we did not find any colorcode with enough color-dimensions that is also valid for color-blindness.

*At a first sight it looks incorrect to say that M, at variance with M_v, can not distinguish between elliptic and hyperbolic areas, since in any plot of M one can clearly identify them. But after some thinking I recognize that there is a real advantage (perhaps the only one) of M vs M_v, which is the fact that ellipticity and hyperbolicity are simply assessed by the maximum or minimum character of M_v, much more easy to automatize that the more complex neighbourhood exploration needed for the case of M. But then I do not understand (and the authors do not give any hint of it) why in Section 4 they say they need a combination of M_v and M, instead of just M_v.*

As the reviewer pointed out, the possible distinction between elliptic and hyperbolic points (more general distinguished hyperbolic trajectories and distinguished trajectories surrounded by an elliptic region in the sense of Mancho et al. (2013)) is the most important property of M_v to make the detection of eddy cores  in flows easier, since one does not need an

additional criterion to discern distinguished hyperbolic trajectories and distinguished trajectories surrounded by an elliptic region in the sense of Mancho et al. (2013) as if one would use M.

For the detection of the eddy shape we have used previously a combination of M and M_v because M shows in our test case a clear line of minimum M values that was easier to detect automated than the line in M_v. In general manifolds correspond to singular lines of M and M_v(Mancho et. al. 2013). To construct an eddy shape detection that is more general and only based on M_v, we have improved the shape detection. The improved shape detection relies on the assumption that the eddy boundary is the largest closed contourline of M_v where M_v is an extremum (large gradient of M_v). We have now rewritten the text accordingly and use now only M_v for the detection of the shape.

*I think that the most original part of this research is the assessment of the behaviour of the different indicators under different types of noise. Nevertheless, the definitions of noise types in page 11 are all incomplete: for type 1 and 2 one can not reproduce the paper results unless the authors define 'noise strength', given that for white noise this would depend on the particular spatial and temporal discretizations used, which are not completely stated. For type 3, it is only after reading a comment in the Supplemental material that one begins to understand that noise is added to the functions h1 and h2, but again, 'strength' or 'noise level' should be properly defined.*

We have rewritten the definition of the noise to clarify how the noise is applied and what we take as a noise strength.

*In the Supplemental material, Sect. S1 there is no indication on how the time-dependence, needed to define T_c, is introduced in the seeded eddy model. Also I find very convoluted (and not well explained) the way the radii of the eddies are sampled. Since at the end the authors restrict to 15-25 km radii, it seems to me that all this complexity is irrelevant and that anything uniform or Gaussian in that range will give the same results.*

The eddies in the seeded eddy model live infinite (which is not very realistic). We have chosen this setup because we were only interested in the detection of the different eddy shapes. The way of choosing the eddy radii was due to the fact that we would like to use the same distribution of radii as Abraham (1998) but restrict ourselves to a specific part of the distribution. We agree that one can even use a simpler distribution, because the model is already very artificial and simplified. In the revised version with have changed that completely: We have removed this rather artificial example of the eddy seeded model and replaced it by an example of a velocity field of the western Baltic Sea according to the suggestions of reviewer #2.

*Errata: * There is a missing square of the velocity in Eq. (3) * Page 12, line 14: signal to noise ratio small? or large?*

Thank you very much for pointing to those typos. We have made the corresponding corrections.

---

## Author Comment (AC4) · 18 May 2016

Response to Comment #1, #2 and the editor comment:

Comment #1: We would like to thank George Haller for pointing out this manuscript to us, which we were indeed not aware of. We first got to know that he and his group are working on similar things as us on a workshop in Potsdam, were the first author of the manuscript (R.V.-K.) presented a poster about the content of this manuscript. The method suggested by Haller et al. 2016 is very similar to ours, which has been developed independently based on a different concept. There are several differences between the two methods. Each of them has still difficulties in finding all eddies, which are partly short lived. Haller et al. 2016 show with their examples that one can indeed nicely identify eddies in ocean flows, which are large and long lived. However, for many applications the real challenge is to identify and track small and short-lived eddies. Our method is a contribution in this direction and as such a step forward towards finding robust methods dealing with the census of eddies.

In the revised version we have included the analysis of an oceanographic flow in the western Baltic Sea. We computed all eddies including their shapes and compare our results to the ones obtained by an eddy tracking tool box by Nencioli et al. (2010). This example reveals the advantages and disadvantages of both methods.

Comment #2 and editors comment: The main problem, which has been addressed by the second comment and answered partly by the editor comment, is the problem of objectivity. This discussion is going on in the community for a long time and is reflected by the comments and the reviewer #1 suggestions. We would like to emphasize that objectivity is not the main focus of our manuscript and it is not our goal to resolve this long-lasting debate. Our goal is to provide oceanographers with a tool, which makes it possible to do a reasonable census of vortices in oceanographic flow fields. This can be provided by several methods and we have added another one which is in fact the same as proposed by Mancho et al. (2013) but using another quantity for the construction of the Lagrangian descriptor.

Objectivity is an important mathematical property as pointed out by George Haller, since it allows to define Lagrangian coherent structures independent of the frame. But in her answer Ana Mancho discusses, that one has of course to expect that a coordinate transformation would change M, but would still give the right answer in the coordinate system used (cf. frame invariance section in Mendoza and Mancho (2012)). Therefore, we think that both approaches are valid in the context of their framework and one has to see them in relation to the coordinate system in which they are computed. The same applies in principle to local Lyapunov exponents as another method to identify Lagrangian coherent structures. They are also not invariant with respect to coordinate transformations. After a coordinate transformation one would get other local or finite time Lyapunov exponents, which still give reasonable answers to the original problem. Only the long-term Lyapunov exponents computed as time goes to infinity are unique characteristics of a chaotic process.

From the application point of view, it is in many cases important, that a certain suitable coordinate system is used to analyze a problem in this particular coordinate system. This is for instance true for oceanographic problems which are given in the earth's coordinates where e.g. coastlines are well defined boundary conditions. Talking to oceanographers it turns out that they find objectivity of being of secondary importance since the coordinate system of the earth is given and rotations of this coordinate system means e.g. rotations of coastlines, which might not be useful in many contexts. When identifying eddies in ocean flows one always has to deal with those boundary conditions since the computation of trajectories of particles contains the problem that particles reach those boundaries. One has to solve this problem by either reflecting them or losing them for the rest of the computation. This poses an additional problem which has not been addressed in many algorithms.

The submitted manuscript does not claim to solve this problem of objectivity and this is not at all the aim of this submission. The aim is to step forward in providing suitable tools for the census of eddies, a problem, which is of increasing interest in oceanography. None of the tools we have checked including ours so far are good enough to solve those problems in oceanography, they all have their pro's and con's, they might be frame-dependent or not. The deficiencies become even more pronounced when looking at data, which are corrupted by large noise. Despite of that one still wants to get some reasonable results on the lifetime, the tracks and the shape of vortices. To solve this problem is a task for the whole community working on the identification of Lagrangian coherent structures.

---

## Author Response (AR1)

To complete the point-to-point responses to the reviews in the author comments AC1, AC2 and AC3 and to the short comments in AC4, a marked-up version of the manuscript and the supplemental material is provided in the following.

List of relevant major changes in the manuscript:

- Change of the title of the paper to avoid the misleading phrase Euler-Lagrangian descriptor (We have replaced the phrase Euler-Lagrangian descriptor in the whole manuscript by Lagrangian descriptor $M_V$ based on the modulus of vorticity)
- Correction of misleading statements in the introduction (pp. 2-3 in the marked up version) as well as including the references pointed out by the referees
- Extended explanation of the Lagrangian descriptors M and $M_V$ and their features (Sect. 2 pp. 5-7 in the marked-up version)
- Change of the colorcode Fig. 1, Fig. 2, Fig. 3, Fig. 4, Fig. 5, Fig. 7, Fig. 8 and Fig. 9
- Extended explanation the idea of the eddy and eddy tracking based on $M_V$ (p. 8 line 16 to p. 10 line 13 in the marked-up version)
- Extended discussion of the choice of $\tau$ including an improved Fig. 6 (p. 12 line 1 to p . 13 line 3 in the marked-up version)
- Partly rewritten Sect. 4.2 to clarify the concept of noise and what is aimed with the idea to add noise to the velocity field (Sect. 4.2 pp. 13-17 in the marked-up version)
- Rewritten Sect. 4.3 including a revised version of the eddy shape detection only based on $M_V$ and an example of the eddy shapes for the western Baltic Sea instead of the seeded eddy model example (Sect. 4.3 pp. 17-22)
- New Fig. 10 showing the results of the eddy shape detection based on $M_V$
- New Fig. 11 showing eddy shapes for an example in the western Baltic Sea
- Rewritten Sect. 5 Discussion and conclusion explaining the advantages and disadvantages of the Lagrangian descriptor $M_V$ and the eddy tracking based on it
- Minor changes in phrasing, grammar, spelling and punctuation in the whole manuscript as well as the correction of equation 3 (p. 5 line 20 in the marked-up version)

List of the relevant major changes in the supplemental material:

- Deleted Sect. S1 about the seeded eddy model, because we have replaced the example in the manuscript (p. 1 in the marked-up version)
- Extended Section about the algorithm of the eddy tracking. Now it includes an explanation of the eddy shape detection based on $M_V$ and a new Fig. S3 (pp. 2-6 in the marked-up version)
- Minor changes in phrasing, grammar, spelling and punctuation in the whole supplemental material

[revised manuscript text omitted]
. The eddy shape detection searches for the largest closed contourline with the largest gradient of $M_V$ along the contour line. To maximise these two conditions at the same time, we maximise a quantity that combines this two ideas: (Area enclosed by the contour line)·(($\sum$ gradient of $M_V$ along contour line)/(length of contour line)). Because contour lines surround the eddy core like the layers of an onion, maximising the enclosed area includes maximising the length of the closed contourline. Maximising the gradient of

10   $M_V$ along the contourline is linked to the idea to search for a singular line (the eddy boundary).

In case of eddy shape detection for realistic oceanic velocity fields like the example of the western Baltic Sea, the coordinates of the eddy cores have to be understood as candidates for the eddy core.

From all candidates for eddy cores only those are kept which fulfil the following conditions:

- The convexity deficiency as defined in Haller et al. (2016) has to be smaller than a threshold.

15 - No land is enclosed in the contourline.

- The contourlines are longer than a threshold length. The reason for that criterion is mainly to speed up the computation. Very short contourlines can typically be found very close around the eddy cores. Therefore, they need not to be checked for the gradient of $M_V$, because they will not describe the eddy boundary.

This set of eddies can then be used as input for eddy tracking.

[Figure]

**Figure S1.** Schematic sketch of the eddy detection algorithm based on  Lagrangian descriptor $M_V$.

[Figure]

**Figure S2.** Schematic sketch of the eddy tracking algorithm based on  Lagrangian descriptor $M_V$.

Sandulescu, M., Hernández-García, E., López, C., and Feudel, U.: Kinematic studies of transport across an island wake, with application to Canary islands, Tellus A, 58, 605–615, 2006.

[Figure]

**Figure S3.** Schematic sketch of the eddy shape algorithm based on the  Lagrangian descriptor $\cancel{M}\underset{\sim}{M_V}$.